# Impact on the stratocumulus-to-cumulus transition of the interaction of cloud microphysics and macrophysics with large-scale circulation

Je-Yun Chun[1], Robert Wood[1], Peter N. Blossey[1], and Sarah J. Doherty[1,2]

[1]Department of Atmospheric Sciences, University of Washington, Seattle, USA
[2]Cooperative Institute for Climate, Ocean and Ecosystems Studies, University of Washington, Seattle, USA

**Correspondence:** Robert Wood (robwood2@uw.edu)

**Abstract.**

This study examines the impact of the interaction of cloud microphysics and macrophysics with the large-scale circulation on stratocumulus-to-cumulus transition (SCT) by combining large-eddy simulation (LES) with a parameterization of weak temperature gradient (WTG) stratified adjustment. The WTG approximates the interaction with the large-scale circulation by inducing domain-mean subsidence to compensate for buoyancy perturbations with respect to a reference thermodynamic profile. A stationary sea-salt sprayer perturbs the transitioning clouds over the Lagrangian domain moving along the trajectory. It is revealed that the cloud response to aerosol perturbation is markedly different depending on whether stratified adjustments in the large-scale circulation in response to buoyancy perturbations are considered. In both cases, aerosol injection into heavily precipitating clouds suppresses precipitation and enhances entrainment. Without application of WTG, cloud-top height rises without a compensating adjustment in subsidence, and the drizzle-induced thinning of the stratocumulus layer is delayed by several days. When WTG adjustment is applied, intensified large-scale subsidence restrains the growth of cloud top height, and increases warming and drying of the stratocumulus layer leads to cloud thinning. The thinned clouds, characterized by reduced emissivity and weakened longwave (LW) radiative cooling efficiency, become more susceptible to cloud breakup. Simultaneously, the reduced sensible heat flux from the surface by precipitation suppression reduces turbulence within the boundary layer. For lightly precipitating clouds, the transition, mainly driven by the warming effect due to enhanced entrainment by increased sea-surface temperature ('deepening-warming' mechanism), is hastened by aerosol injection due to accelerated cloud thinning. For heavily precipitating stratocumulus, in which the pace of SCT is fast due to the loss of clouds by drizzle ('drizzle-depletion' feedback), aerosol injection delays the transition by only a few hours because the deepening-warming mechanism becomes more important by intensified subsidence. Our results imply that the magnitude of the cooling effects of aerosol may be overestimated by as much as $\sim$15-30 $\mathrm{W\,m^{-2}}$ when the adjustment in large-scale circulation is not accounted for in a limited-domain model simulations.

# 1 Introduction

Subtropical marine clouds are a focal area in climate research due to their pivotal role in shaping the Earth's energy balance (Wood, 2012). The cooling influence of overcast low marine clouds results from their efficient reflection of solar insolation and by their emission of outgoing longwave radiative flux being comparable in intensity to that from the ocean surface. As air masses flow equatorward along the trade winds, these clouds undergo cloud breakup within a few days, a process referred to as the stratocumulus-to-cumulus transition (SCT). The SCT is driven by rising sea surface temperatures (SSTs) and evolving meteorological dynamics associated with the descending branches of the Hadley circulation (e.g., Albrecht et al., 1995; Norris, 1998; Wood and Bretherton, 2006). The large-scale circulation modulates large-scale subsidence (e.g., Myers and Norris, 2013; van der Dussen et al., 2016) and inversion instability (e.g., Wood and Bretherton, 2006; Sandu et al., 2010), both of which significantly impact the SCT. The large-scale circulation, in turn, responds to variations in the heat balance caused by changes in the meridional gradient of SST (e.g., Bjerknes, 1966) and by radiative forcing. The current generation of weather and climate models have particular difficulty in constraining the radiative effects of clouds along the SCT because the pivotal processes relevant to the clouds span such a broad range of scales and because interactions between scales are complex (Bony and Dufresne, 2005; Zelinka et al., 2017).

One fundamental mechanism causing the SCT is the 'deepening-warming' feedback (Krueger et al., 1995; Bretherton and Wyant, 1997; Wyant et al., 1997). As air masses are advected with the trade winds over a warming ocean surface, surface latent heat flux (LHF) increases, thereby deepening the marine boundary layer (MBL). Cloud-top cooling, which is a main source of boundary layer turbulence over the shallow stratocumulus-topped MBL, becomes insufficient to overcome warming caused by the entrainment of free-tropospheric air. Simultaneously, the enhanced surface LHF promotes the formulation of cumulus clouds, inducing penetrative entrainment by cumulus updrafts. This further promotes the entrainment of dry and warm air from the free troposphere (FT). The combined effect accelerates the dissipation of the stratocumulus layer, thereby resulting in the SCT. Another process that can drive the SCT is the 'drizzle-depletion' feedback (Wood et al., 2011; Xue et al., 2008; Yamaguchi et al., 2017). Once drizzle is initiated, rain droplets accrete cloud droplets (collision-coalescence, Wood, 2006) and scavenge aerosol particles, both of which ultimately reduce the number of cloud-forming aerosols. The reduced aerosol loading results in larger rain droplets which more effectively collide with other droplets, further reducing aerosol concentrations. This positive feedback loop can quickly deplete the stratocumulus layer, thereby resulting in the SCT.

Aerosols have been identified as an important modulator of cloud and MBL properties, and thus potentially of the SCT and resulting cloud radiative effect. An increase in the concentration of aerosol that act as cloud condensation nuclei results in more numerous and smaller cloud droplets, thereby increasing cloud albedo when cloud macrophysical properties are unchanged (Twomey effect, Twomey, 1974, 1977). This change in cloud droplet size can, however, then influence the cloud macrophysical properties and therefore the SCT. The SCT is delayed and even inhibited when typical pristine low cloud precipitation is suppressed by aerosol perturbations. The reduced cloud droplet and raindrop sizes reduce the collision-coalescence efficiency, thereby interrupting the positive feedback loop of the 'drizzle-depletion' process (Lifetime effect, Albrecht, 1989). On the other hand, with smaller and more numerous cloud droplets the overall droplet sedimentation velocity is reduced, so more droplets

remain closer to the inversion layer (Bretherton et al., 2007). Cloud droplets in contact with the dry and warm free troposphere are efficiently evaporated and, in turn, hasten the mixing of free-tropospheric air into the cloud layer (Zhou et al., 2017). This tends to dessicate the stratocumulus layer, thereby hastening the SCT. Another possible process weakening the aerosol effect of enhancing cloud lifetime is that the cloud deepening caused by drizzle suppression, in turn, increases the potential for rain production enough to offset the initial suppression of precipitation (Stevens and Feingold, 2009; Seifert et al., 2015). These complex, sometimes countervailing cloud responses, are strongly dependent on atmospheric conditions and are entangled with each other (Stevens and Feingold, 2009; Gryspeerdt et al., 2019) such that the net impact on the cloud radiative effect remains very uncertain.

Large-eddy simulation (LES) is a useful tool to investigate the impact of aerosols on SCT because it explicitly resolves processes fundamental to cloud physics and dynamics (e.g., Sandu and Stevens, 2011; Yamaguchi et al., 2017; Blossey et al., 2021). However, it is not yet feasible to routinely run larger regional or global-scale LES simulations due to the huge computational cost. Simulations with more limited (e.g., <100 km) domains, which have been widely used, cannot represent the interactions between all the relevant scales of physics and dynamics ranging from microphysics to the large-scale circulation, all of which play an important role in the SCT. Previous studies have demonstrated that the modification of large-scale thermodynamics and dynamics can modulate the SCT. Diamond et al. (2022) shows that the adjustment in subsidence can modify the thermodynamic profiles as well as the entrainment drying and warming. Dagan (2022) reveals that the large-scale changes in the thermodynamic and dynamic conditions by subtropical rain suppression potentially enhance tropical cloudiness. This interplay between the modification in microphysics and large-scale conditions complicates the response of clouds to aerosol perturbation.

The primary objective of this study is to comprehensively explore the intricate interplay between various scales, spanning microphysics to the large-scale circulation, and their impact on SCT. Within this context, our investigation focuses on evaluating the potential effectiveness and practicality of implementing a climate intervention approach known as Marine Cloud Brightening (MCB) (Latham, 1990). MCB involves the deliberate injection of sea-salt aerosols into subtropical low clouds as a means to counteract anthropogenic global warming.

A question is whether cloud macrophysical changes resulting from aerosol perturbations characteristic of MCB would be affected by the adjustment of the large-scale circulation. Localized perturbations in convection, which can be caused by a strong horizontal gradient of precipitation, significantly modify marine boundary layer buoyancy profiles. Yet, as validated from observations such as in pockets of open cells and ship tracks, horizontal gradients in buoyancy and boundary layer depth are small (Bretherton et al., 2010). This results from an adjustment in subsidence, which acts as a remote feedback that homogenizes the buoyancy profile (Sobel and Bretherton, 2000).

To quantify the effect of this adjustment on cloud evolution, we employ a Large-Eddy Simulation (LES) model integrated with a Weak-Temperature Gradient (WTG) approximation, where the latter parameterizes the remote adjustments of subsidence facilitated by gravity waves, thereby minimizing tropical buoyancy perturbations (Sobel and Bretherton, 2000; Blossey et al., 2009; Bretherton and Blossey, 2017). Aerosol-cloud interactions in deep convective clouds have been studied using WTG methods (e.g., Anber et al., 2019; Abbott and Cronin, 2021). WTG methods have also been extended to model the interactions

and exchanges between two separate simulations (e.g., Daleu et al., 2012), and Dagan (2022) used such a two-column WTG method to study the impact of aerosol perturbations on the coupled evolution of tropical and subtropical columns, finding that subtropical aerosol perturbations could have downstream impacts if they reach deep convective regions. Because our focus on the SCT places our simulations far from deep convective regions and for simplicity, our implementation of WTG includes only a single column (our LES simulation) and uses fixed large-scale buoyancy soundings. Such single-column WTG methods do not, in general, account for the impact of local heating anomalies on the large-scale buoyancy sounding that might result from dense and basin-wide application of MCB. Larger-scale (e.g. regional to global-scale) simulations will explicitly simulate these large-scale circulation responses to aerosol perturbations and cloud brightening. However, these larger-scale models can not resolve the small-scale processes driving cloud adjustments to perturbations, and as such will not capture the coupling between cloud responses and the large-circulation change, and the resulting feedbacks to cloud brightening.

This article is structured as follows: in Section 2, we provide detailed insights into our model configurations and WTG setup, while in Section 3, we present a comparative analysis of simulation outcomes, contrasting scenarios with and without WTG and with and without aerosol injection.

## 2    Methods

### 2.1    Model

For the LES modeling we use the System for Atmospheric Modeling (SAM) version 6.10, built with a finite difference representation of the anelastic system on the Arakawa C-grid spatial discretization (Khairoutdinov and Randall, 2003). The horizontal grid spacing is 50 meters in both the x and y directions, and the vertical resolution is 10 meters near the cloud layer (600-2000 meters altitude), gradually stretching to 15 meters down to the surface (0-600 meters) and up to 100 meters at the top of the model domain (3600 meters). The horizontal domain size is $51.2x12.8x3.6$ km$^3$, with grid numbers 1024x256x216. The model time step is adaptive with a typical value of $\sim$0.5-1 seconds. Doubly periodic boundary conditions are used for both the x and y directions. Advection of scalar fields, such as moisture, liquid-ice static energy, and aerosols, is treated using an advection scheme that preserves monotonicity (Blossey and Durran, 2008). The subgrid-scale turbulence is parameterized using a 1.5-order turbulent closure model with a prognostic formulation of turbulent kinetic energy. The Rapid Radiative Transfer Model for GCM application (RRTMG, Mlawer et al., 1997) calculates short- and long-wave radiative transfer and includes the overlying atmosphere above the LES model top ($\approx$3.6 km) in the computation of radiative heating rates and fluxes. Sensible and latent heat fluxes from the ocean surface in each grid box are estimated considering interaction with the surface turbulence based on Monin–Obukhov theory.

The two-moment Morrison microphysics scheme (Morrison and Grabowski, 2008) predicts the number concentrations and mixing ratios of liquid water cloud droplets and rain droplets. The cloud microphysics scheme is coupled with a bulk aerosol scheme (Berner et al., 2013), predicting the number and dry mass of a single lognormal accumulation mode with a fixed geometric standard deviation of 1.5. The conversion among dry aerosols, clouds, and rain droplets is represented using activation

(Abdul-Razzak and Ghan, 2000), autoconversion and accretion (Khairoutdinov and Kogan, 2000), precipitation evaporation, and scavenging of interstitial aerosol by cloud droplets and rain (see appendix in Berner et al., 2013).

The natural sea-salt spray aerosol number and mass fluxes are diagnosed based on the wind speed and all of these aerosols are in a single, lognormal accumulation mode with a characteristic radius of 130 nm (Clarke et al., 2006; Berner et al., 2013). The injected aerosols also have a mean dry diameter of 255 nm, because the model version used here has a single-mode aerosol scheme with a fixed width, and is not set up to accurately represent aerosol particles with a wide range of sizes. This size is larger than that been identified as the optimal size for the purpose of MCB (30-100 nm in diameter, Wood, 2021).

Condensation occurs when the water vapor mixing ratio exceeds the saturation mixing ratio, which is estimated using saturation adjustment. Thus, only the influence of cloud droplet size and number on droplet sedimentation is considered, while their impact on droplet evaporation is not (see e.g., Ackerman et al., 2004, 2009). Ice-phase hydrometeor species are not considered because the simulation domain is below the freezing level everywhere.

## 2.2   Weak temperature gradient (WTG)

The WTG scheme used in this study is based on the approach given in Appendix A in Blossey et al. (2009). The Weak Temperature Gradient (WTG) approximation assumes that in low-latitude regions, gravity waves quickly adjust the temperature profile in each atmospheric column to achieve a nearly uniform (i.e., minimized horizontal temperature gradient), moist-adiabatic state across the tropics. This adjustment leads to a vertical velocity feedback mechanism, which compensates for diabatic heating differences by inducing large-scale vertical motion. Under WTG, this induced vertical velocity serves to

regulate temperature anomalies by redistributing air within the column. The basic principle of the WTG approximation is that domain-mean anomalies of virtual temperature in the simulated column are calculated relative to location- and time-dependent climatological buoyancy profiles, and these are used as the primary drivers of the perturbation in the column mean vertical motion. This approach ensures that any perturbations in heating within the column prompt an offsetting large-scale subsidence or ascent, maintaining thermal stability.

The core of the WTG approach in this study involves solving a modified gravity wave equation, as presented in Blossey et al. (2009), to adjust vertical velocity in response to column heating. This adjustment can be described by the following equation:

$$\frac{\partial}{\partial p}\left(\frac{f^2 + a_m^2}{a_m}\frac{\partial \omega'}{\partial p}\right) = k^2 \frac{R_d T_v'}{p} \tag{1}$$

where, $\omega'$ is pressure velocity perturbation in response to local temperature anomalies; $f$ is coriolis parameter, representing rotational effects on large-scale circulation; $a_m$ is momentum damping rate, which parameterizes the effect of gravity waves

and turbulence; $k$ is horizontal wavenumber, defining the scale over which temperature anomalies adjust; $R_d$ is specific gas constant for dry air; $T_v'$ is virtual temperature anomaly, representing the deviation of the column temperature from a reference state; $p$ is pressure, with boundaries at surface and tropopause pressures. Here, we use vertically-uniform $a_m = 0.5d^{-1}$, and a value of horizontal wavenumber $k$ corresponding to a wavelength of 1300 km, as in Blossey et al. (2009).

    Sandu et al. (2010) demonstrated that the composite transition derived from Lagrangian analysis closely resembles a clima-

155 tological transition constructed in an Eulerian framework. Based on this, we assume that the climatological buoyancy profiles

along the composite trajectory accurately represent the background buoyancy profiles. For our analysis, we use ERA5 climatological data from the summer months of 2002-2005 [June-August (JJA)] along the climatological trajectory over the Northeast Pacific, as detailed in Table C1 of Sandu et al. (2010). The specifics of how subsidence adjustments are applied using the WTG approach are described in Appendix A of Blossey et al. (2009).

## 2.3 Data

The SCT simulations are based on the initial profiles and SST evolution developed by Sandu and Stevens (2011) and the modified subsidence for a better representation of the subsidence rate at the inversion height in Bretherton and Blossey (2014), which are used for the 3-day Lagrangian advection of a composite climatological, low-level isobaric trajectory over the JJA Northeast Pacific; therefore, the detailed descriptions of the setups are provided there. Here, we mainly focus on the reference (REF) and the fast (FAST) transition cases. The FAST case has a higher SST, weaker large-scale divergence, deeper initial boundary layer depth, and higher moisture in the MBL and FT than the REF case (see Figs. 2 and 5 in Sandu and Stevens, 2011). The potential temperature jump across the inversion layer is much weaker in the FAST than in the REF case.

Table 1 summarizes the cases analyzed in this study. This study mainly focuses on three cases. The first case ($REF_{NO}$) is the REF case without implementation of the WTG, and thus, subsidence is prescribed as given in Bretherton and Blossey (2014). The remaining two cases ($REF_{WTG}$ and $FAST_{WTG}$) are the REF and FAST cases with the implementation of the WTG, respectively, so that subsidence is adjusted based on the ERA5 climatological thermodynamic profiles as described in Section 2.2. Although subsidence correction by Bretherton and Blossey (2014) does reduce buoyancy anomalies, small anomalies can still persist. Consequently, the implementation of WTG induces a minor change in subsidence within the simulation, leading to variations in cloud and MBL properties. This rationale underlies the necessity of running simulations without aerosol injection for both $REF_{NO}$ and $REF_{WTG}$ cases to establish a clear baseline for comparing background and perturbed conditions. The aerosol number concentration, $N_a$, in the MBL and FT are 33 and 100 $cm^{-3}$, respectively, in the three cases. The lower value of MBL $N_a$ (i.e., 33 $cm^{-3}$) than climatological value (greater than $\sim$100 $cm^{-3}$) is chosen to produce precipitation enough to be influenced by aerosol injections, while FT $N_a$ value is a climatological mean value (e.g., Mohrmann et al., 2018). To test the sensitivity to aerosol loadings, three additional REF cases with WTG are conducted: (i) the aerosol injection rate is reduced to one fourth of that in other cases ($REF_{weak}$); (ii) $N_a$ in the lower FT is reduced from 100 to 55 $cm^{-3}$ ($REF_{FT}$); (iii) background $N_a$ in the MBL is increased to 300 $cm^{-3}$ ($REF_{MBL}$). Since the results from the additional three cases are consistent with the conclusions from the main cases, we briefly investigate them in Appendix C.

To allow the MBL and clouds to sufficiently evolve, the runs are spun up for 18 hours nudged to initial profiles with a timescale of 10 minutes. A long spin-up time is chosen to allow the mesoscale organization to fully develop, since it is important for determining cloud adjustments (e.g., through precipitation). Throughout the simulation, temperature and specific humidity 500 m above inversion are nudged with a timescale of one hour to the climatological mean profiles along the trajectory. After the spin-up, the simulations are branched into runs with (PLUME) and without (CTRL) aerosol injection. A stationary point sprayer on the ocean surface injects $1.2 \times 10^{16}$ aerosols per second for $\sim$4.16 hours. This injection rate is for the dry sea salt aerosol and is selected as being sufficient to produce a measurable perturbation in cloud albedo, based on Wood

(2021, their Table 2). Effects on the injected plume buoyancy and mixing due to evaporation of co-emitted water in an actual injected plume are not accounted for, but recent outdoor studies have suggested that this may be less of an issue than has been previously hypothesized (Hernandez-Jaramillo et al., 2024). As noted above, this injected aerosol has a dry diameter of 255 nm. We acknowledge that this is larger than the optimally-sized aerosol for MCB, but as noted above this is necessary to accommodate the model limitation of only being able to accommodate a single aerosol size mode.

The domain is rotated to align the geostrophic background wind with the y-axis (i.e., wind in the lower MBL turns with height and reaches a steady direction to nearly the y-axis in the upper MBL) to minimize the advection of plume in the x direction (Figure 1). Due to the Lagrangian framework, the sprayer effectively moves in the negative y-axis direction and, due to the periodic boundary conditions, passes through the domain about five times. This is analogous to a situation in which air masses traverse a region where several stationary sprayers are spaced at intervals of 12 km along their trajectory, similar to

MCB scenarios given in Wood (2021). Because the point sprayer moves through the domain along the background winds (the y-axis), the diffusion of the injected aerosols is mainly in the x direction. The plume of injected aerosol is quickly dispersed and covers the whole domain within 24 hours (Figure 1). Thus, the impact of mesoscale variability of aerosols on the transitioning clouds is not represented on days 2 and 3.

## 3    Results

### 3.1    Overview of the SCT with and without WTG

Figures 2 and 3 illustrate the evolution of the MBL and cloud properties in the $REF_{NO}$, $REF_{WTG}$ and $FAST_{WTG}$ cases for both the CTRL (solid lines) and PLUME (dashed lines) runs. In Subsection 3.1.1, the CTRL runs are described, and Subsection 3.1.2 discusses the response of the MBL and clouds to aerosol injection.

### 3.1.1    Baseline (CTRL) runs

SCT occurs as a response to the increasing SST in all CTRL cases, as indicated by a gradual decrease in cloud cover, $f_c$, throughout the 3-day simulations (Figure 2c). As in the climatology, the MBL depth increases with SST and reaches 1500 m at the end of the simulations (Figure 2a). In $REF_{NO}$, $f_c$ exceeds 50 percent, and the surface precipitation rate ($R_{sfc}$) is lower than $0.5\,\mathrm{mm\,d^{-1}}$ on Day 1 (Figure 2e). As the clouds deepen with the MBL, $R_{sfc}$ increases to $\sim 0.5\,\mathrm{mm\,d^{-1}}$ on Day 3. After the onset of strong precipitation, the supply of the CCN from the natural sea-salt spray becomes lower than the depletion of

cloud droplets by accretion, leading to a decrease in $N_c$ to $10\,\mathrm{cm^{-3}}$, and $f_c$ becomes lower than 30 percent without significant diurnal variation. This indicates that the SCT is mainly driven by the combined effects of drizzle depletion of clouds and the weakening of overturning circulation in the MBL by the evaporation of strong precipitation below the cloud base.

The adjustment in subsidence through the WTG approximation greatly affects the evolution of clouds and the MBL. The SCT in the $REF_{WTG}$ case is slower than in the $REF_{NO}$ case (e.g. Figure 2c). As illustrated in Appendix A, WTG intensifies the

subsidence in order to reduce the buoyancy perturbation with respect to climatology. Since the aerosol number concentration in

the lower FT is higher than in the MBL, when intensified subsidence is accompanied by enhanced entrainment, more CCN are incorporated into the MBL. The stronger supply of CCN reduces the precipitation rate and, thus, delays the SCT. $f_c$ gradually decreases from 90 to 20 percent with an apparent diurnal cycle driven by solar heating. $N_c$ and $N_a$ remain greater than 20 $\mathrm{cm}^{-3}$, and $R_{sfc}$ is lower than $0.2\,\mathrm{mm\,d}^{-1}$ for the first two days, which implies that the cloud breakup is likely to be caused by cloud thinning by entrainment warming and drying rather than strong precipitation (Figure 2e,g). As the MBL deepens further on Day 3, $R_{sfc}$ increases to $\sim 0.4\,\mathrm{mm\,d}^{-1}$, and precipitation starts to contribute to cloud breakup.

In the $\mathrm{FAST_{WTG}}$ case, the pace of the SCT is faster than for the $\mathrm{REF_{WTG}}$ case and comparable to the reference case without WTG ($\mathrm{REF_{NO}}$). Despite the greater supply of FT aerosol due to stronger entrainment in the $\mathrm{FAST_{WTG}}$ case, a moister FT produces sufficient precipitation to hasten the SCT, as indicated by a similar evolution of $f_c$ and $R_{sfc}$. The similarity between $\mathrm{REF_{NO}}$ and $\mathrm{FAST_{WTG}}$ makes analysis of the impact of the adjustment in subsidence on the SCT feasible.

### 3.1.2 Runs with aerosol injections (PLUME)

Aerosol injection quickly elevates aerosol concentrations throughout the MBL. $N_a$ and $N_c$ rapidly increase to 250 and 200 $\mathrm{cm}^{-3}$, respectively, during the period of the injection. Notably, most of the injected aerosols are activated despite being injected during the daytime when boundary layer turbulence weakens. This result runs counter to the argument from Jenkins et al. (2013) that most of the injected aerosols are not activated because the turbulence during daytime is not strong enough to deliver the injected aerosols to the cloud base to activate them. A smaller activation fraction in Jenkins et al. (2013) may also reflect a greater fraction of aerosol smaller than 100 nm used in that study.

For the $\mathrm{REF_{NO}}$ case, aerosol injection induces increased entrainment and rapid growth of the MBL, which raises the cloud-top height (Fig. 2a). On Day 1, $f_c$ becomes close to overcast at night and early the following day, but rapidly decreases to 30 % in the afternoon. On days 2-3, however, despite the deepening and decoupling of the MBL, the cloud-layer motions — intensified by suppression of cloud-base precipitation — still supply enough moisture to maintain a stratocumulus layer below the inversion. This is why the stratocumulus layer is not dissipated, although the $w_e$ enhancement (Fig. 2f) elevates the cloud base by drying and warming the MBL. In response to the increase in cloud number concentration with aerosol injection, $R_{sfc}$ in the $\mathrm{REF_{NO}}$ decreases to $<0.1\,\mathrm{mm\,d}^{-1}$ on Days 1 and 2 (Figure 2e). On Day 3, $R_{sfc}$ becomes greater in the PLUME run than in the CTRL run, because the deeper ($\sim 1500$ meters) cumulus cloud depth (Figure 3d) can produce strong precipitation despite having a higher cloud droplet number concentration. This implies that when the adjustment in subsidence is not considered, the suppression of significant precipitation by aerosol injection is likely to delay the transition from overcast stratocumulus to cumulus for a couple of days (Erfani et al., 2022). We can expect from the significant $R_{sfc}$ on Day 3 that a few days following the aerosol injection, the SCT starts to occur due to a positive feedback loop in which strong precipitation scavenges aerosols, leading to low cloud droplet number concentrations and thus greater precipitation (Yamaguchi et al., 2017).

Allowing subsidence to respond to buoyancy anomalies induced by aerosol perturbations strongly affects the MBL and cloud evolution. As the aerosol injection in the $\mathrm{REF_{WTG}}$ and $\mathrm{FAST_{WTG}}$ cases increases entrainment and induces boundary layer deepening, subsidence intensifies to bring the sounding in the LES model closer to the reference (ERA JJA climatological) sounding along the trajectory. The intensified subsidence from aerosol injection suppresses the MBL deepening in the

255 $REF_{WTG}$ and $FAST_{WTG}$ cases (Figure 2c), consistent with Dagan et al. (2022) who found a similar response to aerosol enhancement in shallow convective clouds. Without an adjustment in subsidence in the $REF_{NO}$ case, the boundary layer depth increases by 700 m in three days, while those in the $REF_{WTG}$ and $FAST_{WTG}$ deepen only by 100-400 meters. In all three cases, precipitation suppression results in increased turbulence and entrainment.

When subsidence adjustment is accounted for (i.e., $REF_{WTG}$), aerosol injection hastens the SCT compared with $REF_{NO}$.
The combined effects of the enhanced $w_e$ and intensified subsidence induce a much faster cloud thinning, leading to the earlier dissipation of the stratocumulus layer (Figure 3e). Even in the $FAST_{WTG}$ case, where precipitation is significantly suppressed, and thus LWP and $f_c$ increase, the stratocumulus layer completely dissipates within the three-day simulation. This indicates that accounting for the subsidence adjustment has a major effect on the cloud evolution and, thus, the timing of the SCT.

One notable feature of the effect on cloud evolution of aerosol injection in the $REF_{WTG}$ and $FAST_{WTG}$ cases is a rapid
decrease in $f_c$ and in LWP in the afternoon (Figure 2a). Even in the $FAST_{WTG}$ case, where precipitation is significantly suppressed, the $f_c$ and LWP in the PLUME case become smaller than in the CTRL case on the second afternoon. Since reflection of solar SW radiation only occurs during daytime, this diurnal variation should significantly account for the variation in the cloud radiative effect. In the $REF_{NO}$ case, on the other hand, after Day 1.5 $f_c$ and LWP are always greater in the PLUME run than in the CTRL run. In Section 3.2 and Appendix B, the characteristics of boundary-layer turbulence are investigated to
more deeply understand the impact of the adjustment in subsidence and the aerosol perturbation on the SCT.

### 3.2 Boundary layer turbulence

In the MBL, the main sources of turbulence are cloud-top cooling, surface buoyancy flux, and latent heating, which in decoupled or cumulus-coupled boundary layers, occurs in cumulus updrafts. Cloud-top cooling dominates the stratocumulus regime due to the high cloud cover and low SSTs, while surface buoyancy flux and latent heating in Cu updrafts dominate in the
275 cumulus regime due to low cloud cover and high SSTs. Figure B1 demonstrates the relationship between cloud radiative heating rate and cloud thickness, as measured by cloud LWP. LW radiative cooling linearly increases with cloud depth below a threshold (in-cloud LWP$\sim 20\,\mathrm{g m^{-2}}$), likely due in part to lower $f_c$ associated with smaller LWP values, and then saturates above it (Figure B1b). On the other hand, SW radiative heating sub-linearly increases with LWP (Figure B1c). Appendix B discusses how the cloud thickness and subsidence rate can influence cloud breakup. The surface buoyancy flux is determined by
280 sensible and latent heat fluxes (hereafter defined as SHF and LHF, respectively), which can be quantified by $B_0 = \overline{w'b'}_{z=0+} = SHF + 0.61\frac{c_p\overline{T}}{L}LHF$. Since the order of magnitude of changes in SHF and LHF from the aerosol injections are similar and $0.61\frac{c_p\overline{T}}{L} \sim 0.07$, changes in SHF play a leading role in changes in the surface buoyancy flux in these simulations.

Figure 4 shows the daily-averaged cloud radiative heating in the upper MBL. The SW cloud radiative heating and LW cooling by at the upper MBL, $R_{SW}^{up}$ and $R_{LW}^{up}$, are calculated by averaging the cloudy-sky (all-sky minus clear-sky) SW and
285 LW radiative heating rates, respectively, in the upper half of the boundary layer. The LW emission from the cloud top to space induces net cooling in the upper MBL, which is partially offset by solar SW absorption. Since the area of clouds with in-cloud LWP exceeding $20\,\mathrm{g m^{-2}}$ becomes smaller with the SCT, $R_{net}^{up}$ also weakens with time (i.e., along the trajectory). The

weakening of $R_{net}^{up}$ along the trajectory in the CTRL runs is less rapid in $\text{REF}_{\text{WTG}}$ than in $\text{REF}_{\text{NO}}$ and $\text{FAST}_{\text{WTG}}$, due to the slower cloud breakup.

Changes in $R_{net}^{up}$ in response to aerosol injections greatly differ depending on whether the WTG adjustment is implemented. When the adjustment is not included, the radiative cooling of the cloud top in $\text{REF}_{\text{NO}}$ becomes extremely effective in response to aerosol injection, and it weakens quite slowly with time (Fig. 4a). This is why the stratocumulus layer is not dissipated, although the MBL is deeper than 2000 m and the surface buoyancy production is smaller in the PLUME run. In the $\text{REF}_{\text{WTG}}$ (Fig. 4b) and $\text{FAST}_{\text{WTG}}$ (Fig. 4c) cases, on the other hand, the net cooling by clouds in the PLUME runs decreases more rapidly along the trajectory. This can largely be attributed to the thinning of clouds under intensified subsidence with consequently smaller outgoing LW emissions. In $\text{REF}_{\text{WTG}}$, $R_{net}^{up}$ is stronger on Day 1 in the PLUME run but weakens after this relative to the CTRL run. This implies that the aerosol perturbation enhances cloud cover on the first day, but the polluted clouds break up more easily afterward. In the $\text{FAST}_{\text{WTG}}$ case, the significant suppression of drizzle with aerosol injection in turn significantly enhances LWP and $f_c$. While this effect also weakens with time, it is more persistent in the $\text{FAST}_{\text{WTG}}$ case than in $\text{REF}_{\text{WTG}}$, possibly because drizzle-depletion plays a stronger role in the SCT in the $\text{FAST}_{\text{WTG}}$ CTRL simulation.

The diurnally-averaged differences in the entrainment rate ($dw_e = w_{e,PLUME} - w_{e,CTRL}$), surface buoyancy flux ($dB_0 = B_{0,PLUME} - B_{0,CTRL}$) and net radiative heating by clouds in the upper MBL ($dR_{net}^{up} = R_{net,PLUME}^{up} - R_{net,CTRL}^{up}$) are shown in Figure 5. The bars for each value indicate the interquartile range of the differences. In all cases, the surface buoyancy flux decreases in the PLUME runs due to drizzle suppression, leading to the reduced evaporative cooling of drizzle in the sub-cloud layer (Figure 5a). On Day 1, $dB_0$ becomes increasingly negative as the plume track spreads, then is unchanged or becomes slightly less negative on Days 2-3 once the plume track covers the whole domain. Notable is that variation on Days 2 and 3 (i.e., the interquartile range) is marginal, indicating that the change in $dB_0$ is robust throughout the whole day. The enhanced entrainment rate makes the lowest layer drier, intensifying the LHF. However, the contribution of entrainment drying and warming to the surface buoyancy flux is negligible, as the MBL becomes more decoupled.

Although the values of $dB_0$ and $dR_{net}^{up}$ (Figures 5b and c) do not quantitatively represent their contribution to the MBL turbulence, they qualitatively represent the turbulence changes in the MBL induced by the aerosol perturbation. Among the three cases, the decrease in $B_0$ is greatest for $\text{REF}_{\text{NO}}$. However, the weaker driving of turbulence by the smaller surface buoyancy flux ($dB_0 \sim$-6 $\text{W}\,\text{m}^{-2}$ on Day 3) is more than offset by stronger radiative cooling in the upper part of the MBL ($dR_{net}^{up} \sim$-15 $\text{W}\,\text{m}^{-2}$ on Day 3). As a result, the increased turbulence in the MBL is intense enough to sustain the stratocumulus layer. This implies that turbulence generated by increased cloud-top cooling surpasses turbulence dissipation by the decreased surface buoyancy flux, so the MBL turbulence is intense enough to sustain the stratocumulus layer. In the $\text{REF}_{\text{WTG}}$ case, $dR_{net}^{up}$ even becomes positive after Day 2, indicating that both a decreased surface buoyancy flux and cloud-top cooling dissipate the turbulence in the PLUME run. In the $\text{FAST}_{\text{WTG}}$ run, $dR_{net}^{up}$ is slightly more negative than $dB_0$ on Day 2 but less negative on Day 3, which implies that the MBL turbulence should be much weaker in the PLUME than CTRL runs despite having similar amounts of precipitation suppression to $\text{REF}_{\text{NO}}$.

The entrainment rate is enhanced by the aerosol perturbation due to the more effective evaporation of cloud water lying at the inversion layer. On Day 1, $dw_e$ increases with time as the plume track spreads, and as the cloud adjustments to the aerosol

injection have not yet reached equilibrium ($\sim$1-4 days Schubert et al., 1979; Wood, 2007; Glassmeier et al., 2021). In the REF$_{NO}$ case, where the SCT is inhibited by the aerosol perturbation, the enhanced cloud-top cooling (Fig.4a) intensifies the turbulence at the inversion layer. In the REF$_{WTG}$ and FAST$_{WTG}$ cases, on the other hand, the entrainment enhancement does not increase on Day 2, because the cloud-top cooling is less effective due to cloud breakup. On Day 3, $dw_e$ becomes much weaker, such that the interquartile range of $dw_e$ includes zero. This implies that entrainment enhancement, which is a dominant process for LWP adjustment over stratocumulus cloud decks, becomes less pronounced as the clouds break up.

Figure 6 illustrates the vertical structure of the diurnally-averaged MBL turbulence on each day. The vertical profiles of buoyancy flux $B$ (Figures 6a-c) are consistent with the findings from Figs. 4 and 5. In general, $B$ is weaker in the subcloud layer in the PLUME runs than in the CTRL runs due to the decrease in $B_0$. In particular, a more negative $B$ at the cloud base in the PLUME run indicates that the MBL becomes more decoupled, driven by enhanced $w_e$. In the cloud layer, $B$ is greater in the PLUME runs than in the CTRL runs because of the stronger radiative cooling associated with greater cloud cover due to rain suppression by the aerosol perturbation. It is notable that $B$ in the cloud layer does not decrease with time in the REF$_{NO}$ case, but in the REF$_{WTG}$ and FAST$_{WTG}$ cases it decreases on Days 2-3 due to the SCT, as illustrated in Fig. 5c.

Figures 6d-f show the vertical profile of vertical-velocity skewness, $S_w = \overline{w'^3}/(\overline{w'^2})^{3/2}$. In subtropical, stratocumulus-capped marine boundary layers, negative $S_w$ generally indicates that turbulence and convection are dominated by downdrafts associated with cloud top cooling, while positive $S_w$ is related to cumulus convection and/or subcloud layer turbulence driven by positive surface buoyancy fluxes ($B_0$). In the CTRL runs, $S_w$ is positive throughout the simulation, due to significant $B_0$, which increases along the trajectory as SST increases. When aerosols are injected into drizzling clouds, $S_w$ at the cloud layer decreases due to smaller $B_0$, the suppression of cloud-base precipitation, and with the increase in cloud-layer turbulence induced by stronger cloud-top cooling. This indicates that the contribution of cloud-top cooling to MBL turbulence is greater in a polluted than in a pristine MBL. On Day 3 (Figure 6f), the decreases in $S_w$ at the cloud layer with aerosol injection become less significant. This is mainly attributed to a weaker turbulence production at cloud-top (Figure 6c) and an increase of SST and cumulus convection with time.

To understand the diurnal variation of clouds in the PLUME and CTRL runs shown in Section 3.1.1, turbulence properties in the night, morning, and afternoon are shown in Figs. 7 and 8. In all cases, buoyancy production in the near-inversion cloud layer (z/z$_{inv}$ $\sim$0.8) is strongest at night, due to the absence of shortwave (SW) solar absorption and the significant longwave (LW) radiative cooling (Figures 7 and 8a). Near-inversion buoyancy production is greater in the PLUME runs than CTRL runs, since the aerosol perturbation's impact on drizzle suppression (resulting in a positive LWP adjustment) is more significant than its impact on entrainment enhancement (negative LWP adjustment). Before noon (Morning), clouds exert a consistent LW radiative cooling since $f_c$ is still high, but the cloud-top cooling effect starts to be partly offset by SW solar absorption (Figures 7 and 8b).

In the afternoon, LW radiative cooling and SW solar absorption both weaken as $f_c$ decreases. Changes in $R_{SW}^{up}$ and $R_{LW}^{up}$ roughly offset each other in the PLUME run for the REF$_{NO}$ case, so the net cooling rate does not vary during the daytime (Figure 7a) and the buoyancy production in the near-inversion cloud layer remains significant (Figure 8c). In the PLUME runs for the REF$_{WTG}$ and FAST$_{WTG}$ cases, however, $R_{LW}^{up}$ significantly decreases so that the net cooling rate in the afternoon

is much smaller than in the morning. The significant decrease in $R^{up}$ (and thus weak buoyancy production at cloud layer) is mainly attributed to the cloud thinning and breakup, with LWP in an increasing fraction of falling below $20\,\mathrm{g\,m^{-2}}$, due to the
impacts of the adjustment in subsidence. The greater $S_w$ within the cloud layer also suggests a complete breakup of Sc layers (Figures 8f). The analysis of diurnal variation in turbulence implies that the application of the WTG method has a pronounced impact on Sc layers during daytime.

### 3.3   Cloud Radiative Effect (CRE)

Previous sections illustrate the adjustments in cloud properties that result from aerosol injections. To analyze the radiative
effect of aerosol perturbations over the various cases, we quantify the changes in the cloud radiative effect (computed at top of atmosphere) caused by changes in cloud properties. Here $\mathrm{dCRE} = \mathrm{CRE_{PLUME}} - \mathrm{CRE_{CTRL}}$ so that positive dCRE indicates a net warming effect, and negative a net cooling effect. The SW cloud radiative effect is decomposed into the components caused by changes in the cloud droplet number concentration ($\mathrm{dCRE_{N_c}}$), cloud thickness ($\mathrm{dCRE_{LWP}}$) and cloud cover ($\mathrm{dCRE_{f_c}}$), as illustrated in Appendix B in Chun et al. (2023). Here, we also consider the LW cloud radiative effect ($\mathrm{dCRE_{LW}}$) since changes
in $f_c$ and inversion height by aerosol injection are not negligible through the SCT.

Table 2 and Figure 9 summarize the decomposed dCRE on each day. As expected, the increase in $N_c$ enhances the cloud albedo throughout the 3-day simulations in all the cases. On Day 1, the injected aerosols are quickly activated to cloud droplets, but the plume track is still narrow. As the plume quickly fills the domain during the first 20 hours and the decrease in domain-mean $N_c$ is slow, the negative $\mathrm{dCRE_{N_c}}$ does not significantly decrease in the $\mathrm{REF_{NO}}$ and $\mathrm{FAST_{WTG}}$ cases, and even increases
in the $\mathrm{REF_{WTG}}$ case, due to a slow decrease in $f_c$. On Day 3, decreases in the magnitude of $\mathrm{dCRE_{N_c}}$ occur in all the cases because of the combined effects of decreased $f_c$ and $N_c$ in the PLUME runs on Day 3.

There is a significant difference in $\mathrm{dCRE_{f_c}}$ between the cases with and without the WTG adjustment. In the $\mathrm{REF_{NO}}$ case, $\mathrm{dCRE_{f_c}}$ is -10.8 $\mathrm{W\,m^{-2}}$ and the cooling effect increases four-fold on Days 2 and 3 (-44.1 and -46.2 $\mathrm{W\,m^{-2}}$, respectively) because the SCT is inhibited. In the $\mathrm{REF_{WTG}}$ case, $\mathrm{dCRE_{f_c}}$ on Day 1 is negative (-2.6 $\mathrm{W\,m^{-2}}$), but becomes positive on
Day 2 (1.8 $\mathrm{W\,m^{-2}}$) and increases to 10.4 $\mathrm{W\,m^{-2}}$ on Day 3. In the $\mathrm{FAST_{WTG}}$, the negative $\mathrm{dCRE_{f_c}}$ on Day 1 (-12.8 $\mathrm{W\,m^{-2}}$), caused by a delayed SCT, becomes more negative on Day 2 (-30.3 $\mathrm{W\,m^{-2}}$). On Day 3, the magnitude of $\mathrm{dCRE_{f_c}}$ is smaller (-10.6 $\mathrm{W\,m^{-2}}$) due to the SCT in the PLUME run. The radiative effect of changes in cloud thickness (i.e., $\mathrm{dCRE_{LWP}}$) is minor on Day 1 in the three cases but becomes more pronounced on Day 2. On Day 3, $\mathrm{dCRE_{LWP}}$ weakens, suggesting that the time scale of adjustment in cloud thickness is roughly one day, as pointed out by Glassmeier et al. (2021). This timescale is
also consistent with the observed timescale for the relaxation of LWP fluctuations in the subtropical regions where SCT occurs (Eastman et al., 2016).

The change in outgoing LW radiative flux, which has not received as much attention in previous studies of low clouds' radiative effect, is nontrivial when the background precipitation is significantly suppressed, and cloud cover increases in response. Since the MBL depths increase along the trajectory, the difference between cloud-top temperature (CTT) and SST
increases. Thus, changes in cloud cover exert a nontrivial change in the LW cloud radiative effect. In the $\mathrm{REF_{NO}}$ case, as the SCT is completely inhibited during the 3-day simulation and the MBL depth rapidly increases in the PLUME run, $\mathrm{dCRE_{LW}}$

increases along the trajectory (from 2.8 $\mathrm{W\,m^{-2}}$ on Day 1 to 17.1 $\mathrm{W\,m^{-2}}$ on Day 3). The magnitude of $\mathrm{dCRE_{LW}}$ is smaller in the $\mathrm{FAST_{WTG}}$ case than in the $\mathrm{REF_{NO}}$, because the MBL deepening is suppressed by intensified subsidence. In the weakly precipitating MBL of the ($\mathrm{REF_{WTG}}$) case, $\mathrm{dCRE_{LW}}$ does not account for the total $\mathrm{dCRE}$, since the decrease in $f_c$ during daytime is compensated at night and CTT does not significantly change due to the adjustment in subsidence.

The total $\mathrm{dCRE}$ (i.e., $\mathrm{dCRE}$ averaged across the 3-day simulations) differs greatly between the cases with and without WTG. In the $\mathrm{REF_{NO}}$ case, a significant total cooling effect (-38.9 $\mathrm{W\,m^{-2}}$) results from roughly equal contributions from the Twomey effect (-17.3 $\mathrm{W\,m^{-2}}$) and changes in cloud macrophysics (-21.6 $\mathrm{W\,m^{-2}}$), associated mostly with the delay in SCT. In the $\mathrm{REF_{WTG}}$ case, the slightly larger $\mathrm{dCRE_{Nc}}$ (-20.5 $\mathrm{W\,m^{-2}}$) is partially offset, by 58 percent, by the change in cloud macrophysics (together, 11.9 $\mathrm{W\,m^{-2}}$), resulting in a smaller, but still significant, net cooling effect of -8.5 $\mathrm{W\,m^{-2}}$. In the $\mathrm{FAST_{WTG}}$ case, a smaller decrease in cloud LWP and an increase in cloud cover augment the negative CRE from the increase in $\mathrm{dCRE_{Nc}}$. Here, the CRE from the change in cloud macrophysical properties (-10.3 $\mathrm{W\,m^{-2}}$) is three quarters of $\mathrm{dCRE_{Nc}}$ (-14.2 $\mathrm{W\,m^{-2}}$). The total $\mathrm{dCRE}$ is -24.6 $\mathrm{W\,m^{-2}}$, which is only 63 percent of that in the $\mathrm{REF_{NO}}$ case, where the change in precipitation is comparable. This implies that if we do not consider the interaction between the aerosol-cloud interactions and the larger-scale circulation, the cloud radiative effect will be overestimated.

## 4 Discussion

One key finding of this study is that incorporating the adjustment in the large-scale circulation to the background thermodynamic state in response to an aerosol perturbation has a crucial impact on the lifetime of stratocumulus clouds. With fixed large-scale dynamics, the aerosol perturbation inhibits a large amount of precipitation, thereby leading to the destabilization and the rapid growth of the boundary layer. Consequently, this inhibits the stratocumulus-to-cumulus transition (SCT). It is anticipated from our results that as marine boundary layer (MBL) continuously deepens with time, or along the trajectory, for a couple of days precipitation is initiated in the clouds deepened through aerosol injection, despite the clouds having a larger number of cloud droplets. This is a type of buffering or deepening effect (Stevens and Feingold, 2009; Seifert et al., 2015), which results in the SCT. With a framework that accounts for an interactive large-scale circulation, on the other hand, intensified subsidence from buoyancy perturbations suppress cloud deepening (Dagan et al., 2022). As a result, the cloud becomes thinner and loses the potential to generate turbulence by cloud-top cooling, inducing cloud breakup. This suggests that the aerosol perturbation does not inhibit the SCT but changes the transition regime from a precipitation-driven 'drizzle-depletion' to an entrainment-driven 'deepening-warming' mechanism.

The stark contrast in the pace of the SCT indicates that the cloud radiative effect will be overestimated if the interplay between the aerosol-cloud interactions and the large-scale circulation is not accounted for. For strongly precipitating MBLs, the reduced longevity of the stratocumulus layer by aerosol perturbation associated with the subsidence intensification reduces the cooling effect to a third compared to cases without the subsidence intensification. For lightly precipitating MBLs, positive LWP adjustments cancel fifty percent of the Twomey effect. This implies that integrating the interplay of aerosol-cloud interactions (ACI) with the large-scale circulation more accurately constrains the radiative forcing associated with anthropogenic climate

change and the deliberate injection of targeted aerosols to mitigate anthropogenic global warming (i.e., climate intervention by marine cloud brightening, MCB).

Our research findings also highlight the contrast in fundamental processes inherent to stratocumulus and cumulus cloud regimes, emphasizing the significance of adopting a regime-centered approach, as recommended by Stevens and Feingold (2009). In the context of overcast shallow stratocumulus, characterized by well-mixed boundary layers, we observe that the intensification of cloud-top entrainment due to an aerosol perturbation drives cloud thinning (i.e., negative cloud adjustments), but that thinning is partly or, in the heavily precipitating state, more than compensated by positive adjustments associated with precipitation suppression (Albrecht, 1989; Ackerman et al., 2004; Wang et al., 2011; Rosenfeld et al., 2019; Chen et al., 2022; Yuan et al., 2023) and increased surface heat fluxes (Chun et al., 2023; Chen et al., 2024). However, as clouds undergo the transition, the influence of cloud-top entrainment adjustments diminishes, owing to a weakened contribution from cloud tops to boundary layer turbulence. Simultaneously, an adjustment in surface buoyancy production becomes more pronounced since the deeper cumulus clouds tend to produce more precipitation than shallower stratocumulus clouds. Given that the impact of subsidence adjustments emerges as the dominant factor when significant changes in precipitation occur, the influence of subsidence adjustments tends to be less pronounced in this regime. In contrast, deeper cumulus clouds exhibit a greater susceptibility to modulation through the interplay between aerosol perturbations and the large-scale circulation.

The realization of the interactive large-scale circulation used in this model, the weak-temperature gradient (WTG), is limited because this method is simplified. For this approximation, the background thermodynamic states for the WTG (e.g., lower free-tropospheric temperature and moisture) do not respond to the background dynamic adjustment. This approximation might be acceptable for 3-day simulations. As the pollution track is widespread, however, the thermodynamic reference state should be changed, thereby affecting the cloud adjustment. In addition, the localized but persistent radiative forcing caused by the aerosol perturbation potentially causes an imbalance in the energy budget, inducing perturbations in the large-scale circulation and thereby, winds, surface fluxes, and clouds (Abbott and Cronin, 2021; Dagan, 2022; Diamond et al., 2022). Another critical point raised by Dagan (2022) is the interaction between different cloud regimes, such as those in the subtropics and tropics. Aerosol loadings in one regime can significantly impact cloud properties in another by perturbing large-scale circulations. This interdependence suggests that understanding the impact of aerosol-cloud interaction on climate system requires a closer investigation of the interconnections across distant regions, where regional aerosol perturbations can propagate through modulating large-scale dynamics to influence cloud and atmospheric properties far from their origin. Such insights underscore the importance of a comprehensive approach in studying climate dynamics, where aerosol-cloud interactions are evaluated in the context of both local processes and their broader, system-wide effects.

The experiments presented in this study offer valuable insights into the impact of aerosol perturbations on the SCT. However, it is important to acknowledge their limitations in fully constraining the radiative effects of aerosol perturbations on the SCT. The analysis primarily focuses on two composite trajectories, the REF and FAST cases derived from Sandu and Stevens (2011), which, while informative, do not fully capture the diverse variability of the boundary layer and cloud behavior during transitions. Additionally, the WTG approach in this study relies on thermodynamic profiles from ERA5 reanalysis, which, though useful, may introduce biases typical of weather and climate models in the SCT regime. ERA5 vertical profiles of tem-

460 perature, moisture, and other variables may not fully capture the subtle thermodynamic gradients and interactions characteristic of SCT regions, potentially affecting the representation of cloud formation and dissipation processes. Moreover, the utilization of a single climatological profile for the WTG in this investigation provides valuable insights but may not comprehensively account for the intricate interactions between cloud microphysics, macrophysics, and large-scale circulations. Therefore, it is imperative for future research to expand upon these findings by conducting experiments representative of various geographic 465 locations where compact low clouds are prevalent, so their breakup processes can be more comprehensively studied.

## 5  Summary

This study explores the interaction of cloud microphysics and macrophysics with large-scale circulation impacts in large-eddy simulations of the stratocumulus-to-cumulus transition (SCT). To account for the interaction of large-scale dynamics with changes in microphysics and macrophysics, we utilized the weak-temperature gradient (WTG) approach. The WTG approach 470 approximates the large-scale dynamical responses to buoyancy perturbations with respect to a reference climatological thermodynamic condition. We investigate two climatological trajectories over the Northeast Pacific (Sandu and Stevens, 2011), where the SCT frequently occurs. These cases are systematically examined through simulations both with and without aerosol injections, providing insights into the intricate response to aerosol perturbations.

Throughout the preceding sections, we have highlighted the pivotal influence of adjustments in the large-scale circulation 475 and its subsequent response to aerosol perturbations on the SCT. The growth of the marine boundary layer (MBL) triggered by aerosol injection introduces a negative buoyancy perturbation, thereby altering the thermodynamic state and inducing enhancement of the subsidence. This intensified subsidence suppresses cloud-top height growth, as observed in previous work (van der Dussen et al., 2016), and simultaneously elevates cloud-base height due to enhanced cloud-top entrainment warming and drying processes. Consequently, this intricate interplay results in an accelerated thinning of the cloud layer.

Due to the intensified subsidence, there is an increase in the fraction of clouds with LWP lower than a critical threshold ($\sim 20\,\mathrm{g\,m^{-2}}$), characterized by a reduced emissivity and weakened longwave radiative cooling efficiency, where clouds become more susceptible to cloud breakup, thereby hastening the SCT. Meanwhile, a reduction in cloud-base precipitation driven by the aerosol perturbation makes the sub-cloud layer warmer, leading to weakened surface buoyancy fluxes. Since cloud depth diurnally decreases with solar absorption and surface buoyancy weakens throughout the day, the cloud breakup driven by the 485 aerosol perturbation becomes more pronounced during daytime.

In a weakly precipitating MBL in which enhanced entrainment drives a 'deepening-warming' transition, the SCT is accelerated due to the decreases in turbulent generation from decreases in both cloud-top cooling and the surface buoyancy flux. In a strongly precipitating MBL in which precipitation drives a 'drizzle-depletion' transition, the cloud amount increases due to the retention of liquid water through suppressed precipitation, but this does not inhibit the SCT. Without accounting for the 490 subsidence adjustment triggered by the aerosol perturbation, the free growth of the MBL through precipitation suppression inhibits the stratocumulus breakup, a phenomenon in alignment with a recent study Prabhakaran et al. (2023).

The MBL and cloud adjustments resulting from the interaction of an aerosol perturbation with the large-scale circulation strongly modulates the cloud radiative effect (CRE). For a lightly drizzling MBL, the Twomey effect brightens the clouds, and this is largely offset by accelerated cloud breakup. Despite the decrease in cloud cover during daytime, the cloud recovery at night makes longwave (LW) radiative forcing a marginal contribution to the total radiative forcing. For a heavily precipitating MBL, a positive LWP adjustment driven by reduced precipitation augments the Twomey brightening, but this enhancement in LWP is smaller than in the case without subsidence adjustment. The increase in cloud cover, especially at night, reduces LW radiative emission into space and partially cancels the SW radiative cooling. Nonetheless, the total change in the cloud radiative effect with aerosol injection exerts a cooling impact across all cases.

## Appendix A:  Adjustment in Large-scale Subsidence by the Weak Temperature Gradient

The inversion height ($z_{inv}$) from the climatology, which represents the large-scale background thermodynamic profiles, is slightly lower than those in the model simulations (Figure 2a). This results in a negative buoyancy perturbation in the upper part of the boundary layer compared to the background. Subsidence is intensified to reduce the negative buoyancy deviation by inducing adiabatic warming (Figure 2b and Figure A1b). The reduction in cloud-base precipitation by an influx of the FT aerosol intensifies in-cloud turbulence, leading to deepening of the MBL. The tendency toward the MBL deepening further intensifies subsidence in order to reduce the virtual potential temperature, $\theta_v$, anomaly in the background state. $\theta_v$ in the FAST$_{\text{WTG}}$ case is higher than in the REF$_{\text{WTG}}$ case due to the higher SST and greater latent heat release by precipitation formation (Figure 2e), leading to weaker subsidence in the FAST$_{\text{WTG}}$ case than in the REF$_{\text{WTG}}$ case. As a result of the subsidence adjustment, the inversion height ($z_{inv}$) in the cases with the WTG remains close to that in the background state. The entrainment rate ($w_e$) is higher in the REF$_{\text{WTG}}$ and FAST$_{\text{WTG}}$ cases than in the REF$_{\text{NO}}$ case, because the deepening rate is comparable, but subsidence is intensified (Figure 2f).

The aerosol injections in the cases with the WTG adjustment (dashed lines in Fig.A1) perturb the buoyancy profile and, thus, the large-scale vertical motion. The decrease in precipitation flux down to the surface with aerosol injection reduces $\theta_v$ in the MBL and, in turn, intensifies subsidence to dampen the decrease in $\theta_v$. In addition, the tendency of the MBL deepening by enhanced $w_e$ leads to an additional negative $\theta_v$ perturbation at the upper part of the boundary layer, resulting in the further intensification of subsidence. The subsidence is intensified further in the FAST$_{\text{WTG}}$ case than in the REF$_{\text{WTG}}$ case due to a greater reduction in precipitation and increased $w_e$. The prescribed feedback loop illustrates that the boundary layer deepening by aerosol enhancement is buffered by intensified subsidence as illustrated in Dagan et al. (2022).

## Appendix B:  Cloud radiative heating rate and MBL collapse as a response to a decrease in cloud thickness

Figure B1 illustrates the dependency of the cloud radiative heating rate on in-cloud LWP in the afternoon. Individual scatter plots indicate radiative heating by clouds in each column. LW radiative cooling sharply increases with cloud depth for in-cloud LWP up to 20 $\text{g m}^{-2}$, then becomes saturated above 20 $\text{g m}^{-2}$. SW radiative heating, on the other hand, sublinearly

increases with LWP. As a result, the net cloud radiative cooling rate sharply strengthens up to a LWP of 20 $\mathrm{g\,m^{-2}}$, then weakens above 20 $\mathrm{g\,m^{-2}}$. As discussed in Bretherton et al. (2010), when the boundary layer depth decreases (i.e., $dz_{inv}/dt = w_e - w_{z=z_{inv}} < 0$, clouds thinner than 20 $\mathrm{g\,m^{-2}}$ quickly dissipate through a positive feedback loop (MBL depth decreases - decrease in cloud thickness - decrease in radiative driving of turbulence and entrainment - decrease in cloud thickness). With the WTG implementation, both weakened turbulence in the MBL (e.g., Figs.4 and 5) and intensified subsidence (Appendix A) by an aerosol perturbation make the stratocumulus more vulnerable to cloud breakup.

## Appendix C: Sensitivity to aerosol conditions

To test the sensitivity of accounting for large-scale circulation responses to aerosol conditions, the $\mathrm{REF_{WTG}}$ case is repeated with three different sets of aerosol conditions, as given in Table 1. The evolution of the cloud and MBL properties are illustrated in Figure C1, similar to Fig.2.

The only difference between the $\mathrm{REF_{weak}}$ and $\mathrm{REF_{WTG}}$ cases is the aerosol injection rate, so the CTRL run is exactly the same as for the $\mathrm{REF_{WTG}}$ case in Fig. 2. In the $\mathrm{REF_{WTG}}$ case PLUME run, $N_a$ in the MBL and $N_c$ are both enhanced to 90 $\mathrm{cm^{-3}}$ – a factor of three smaller aerosol perturbation than in the $\mathrm{REF_{WTG}}$ PLUME case ($\sim$300 $\mathrm{cm^{-3}}$) (Figure C1g,h). This case examines the sensitivity to aerosol perturbation in the SCT with weak precipitation.

In the $\mathrm{REF_{FT}}$ case, the stratocumulus layer has a weaker supply of aerosol from the lower FT. Clouds rapidly break up on the first night due to the rapid scavenging of the larger cloud droplets by stronger precipitation (Figure C1c,e). Due to the stabilization by rain evaporation below the cloud base, $z_{inv}$ is slightly lower than that in the climatology. Due to the polluted MBL, the precipitation rate in the $\mathrm{REF_{MBL}}$ case is too weak to reach the surface ($R_{sfc} < 0.1\ \mathrm{mm\,d^{-1}}$), so the change in $R_{sfc}$ by aerosol injection is negligible. This case examines the impact of aerosol injection on the SCT in a non-precipitating MBL.

With aerosol injections, the clouds and MBL evolve similarly to that in the $\mathrm{REF_{WTG}}$ case. As discussed in Sections 3.1.2 and 3.1.3, the stratocumulus layer gradually thins, with a significant diurnal variation (thinning during the daytime and recovering at night). The decreases in $f_c$ and LWP are comparable in the $\mathrm{REF_{weak}}$ and $\mathrm{REF_{WTG}}$ cases, although $N_c$ enhancement is three times weaker in the former case. This result is consistent with Manshausen et al. (2023), revealing that liquid water increases driven by aerosol perturbations in raining clouds are constant over the emission ranges observed. One discernible feature is a slightly higher $f_c$ and LWP on Day 2 in the $\mathrm{REF_{weak}}$ case than in the $\mathrm{REF_{WTG}}$ case. In the $\mathrm{REF_{FT}}$ case, aerosol injections effectively reduce $R_{sfc}$ and thus increase $f_c$, LWP, $w_e$, and subsidence. The difference in $f_c$ and LWP between the CTRL and PLUME runs is quickly reduced along the trajectory due to the enhanced subsidence, as in the $\mathrm{FAST_{WTG}}$ case. Due to a pristine lower FT, the aerosol number concentration in the MBL and cloud number concentration rapidly decreases with time. In the $\mathrm{REF_{MBL}}$ case, the perturbations caused by aerosol injection are much smaller due to small changes in $w_e$ and $R_{sfc}$.

Changes in cloud radiative effects in the $\mathrm{REF_{weak}}$ case are broadly similar to those in $\mathrm{REF_{WTG}}$ case but weaker in magnitude with an overall dCRE that is negative but about half as strong as in the $\mathrm{REF_{WTG}}$ case. In the $\mathrm{REF_{FT}}$ case, dCRE is more negative than for the $\mathrm{FAST_{WTG}}$ case, due to earlier cloud breakup in the $\mathrm{REF_{FT}}$ case than in the $\mathrm{FAST_{WTG}}$ case. In

the $\text{REF}_{\text{MBL}}$ case, $\text{dCRE}$ on Day 1 is negative, but increases with time and becomes positive on Day 3, leading to a small radiative forcing across all three days.

*Code and data availability.* The original model source code is publicly available: http://rossby.msrc.sunysb.edu/ marat/SAM/ (further information can be found at http://rossby.msrc.sunysb.edu/ marat/SAM.html, Khairoutdinov, 2022). The modified model source codes and case setups for these simulations are available at https://doi.org/10.5281/zenodo.7353468 (Chun, 2022). Python analysis codes for this study are 560 available on request.

*Author contributions.* JYC, RW, and PB formulated the original model study. JYC and PB set up the model runs. JYC conducted and analyzed the runs with input from RW, PB, and SJD. JYC drafted the paper, and RW, PB, and SJD provided edits and revisions.

*Competing interests.* The contact author has declared that none of the authors has any competing interests.

*Acknowledgements.* This study was primarily supported by NOAA's Climate Program Office Earth's Radiation Budget (ERB) Program, 565 Grant NA22OAR4310474, as well as through the University of Washington's Marine Cloud Brightening Program, which is funded by the generous support of a growing consortium of individual and foundation donors. This publication is also partially funded by the Cooperative Institute for Climate, Ocean and Ecosystem Studies (CICOES) under NOAA Cooperative Agreement NA20OAR4320271, Contribution No. 2024-1349. This work used Bridges-2 (Brown et al., 2021) at Pittsburgh Supercomputing Center through allocation EES210037 from the Advanced Cyberinfrastructure Coordination Ecosystem: Services & Support (ACCESS) program (Boerner et al., 2023), which is supported 570 by National Science Foundation Grants 2138259, 2138286, 2138307, 2137603, and 2138296.

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

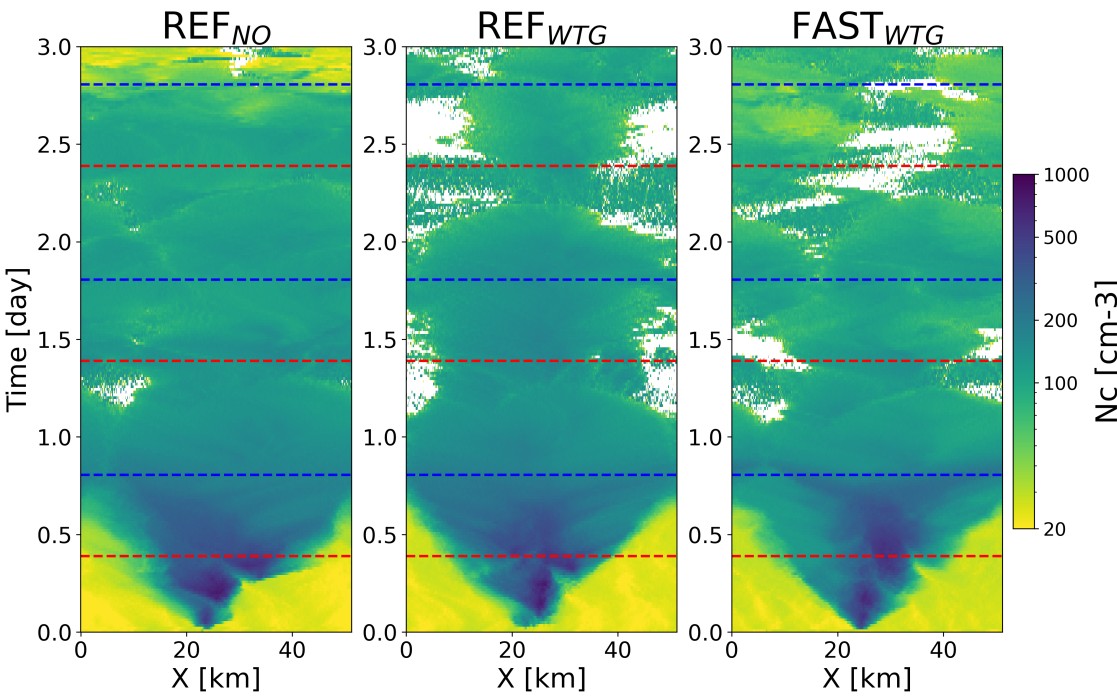

**Figure 1.** Hovṁoller plots of cloud number concentration $N_c$ in the *PLUME* runs for the (a) REF_NO, (b) REF_WTG and (c) FAST_WTG cases, showing the dispersion of the plume in the model domain.

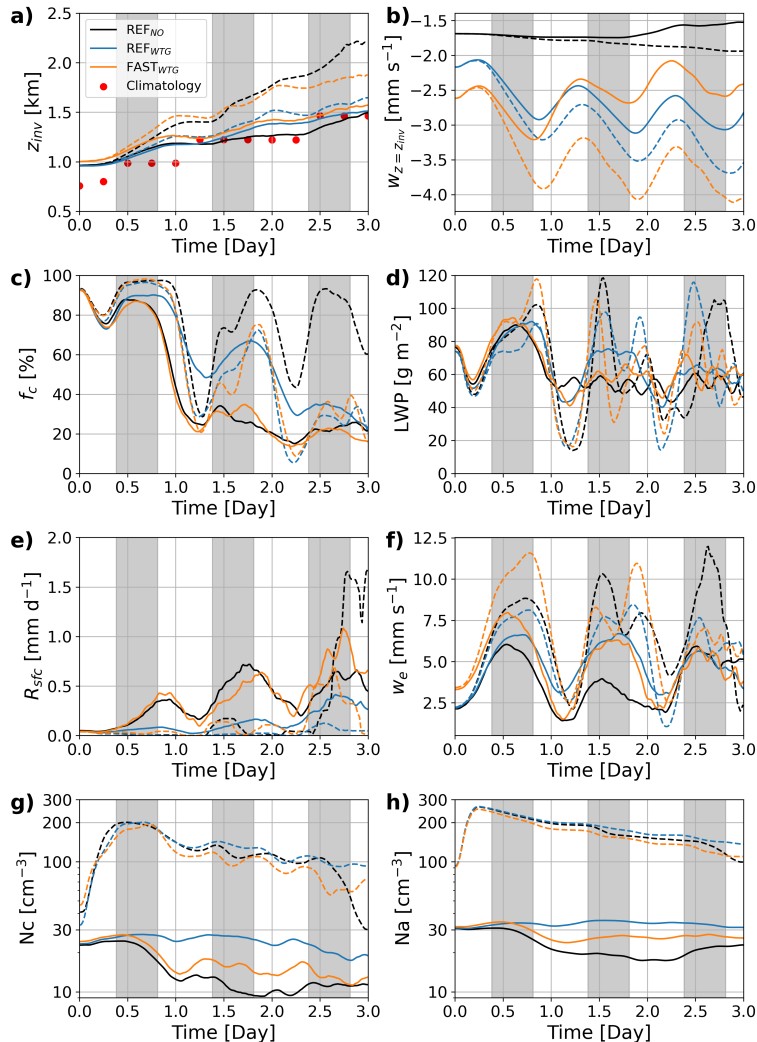

**Figure 2.** Time evolution of cloud and marine boundary layer properties: (a) inversion height, (b) vertical velocity at inversion height, (c) cloud cover ($f_c$), (d) cloud liquid water path (LWP), (e) surface rain rate ($R_{sfc}$), (f) entrainment rate ($w_e$), (g) cloud droplet number concentration, (h) total (cloud+aerosol numbers aerosol number concentration ). The solid lines indicate the values for the CTRL runs and the dashed lines for the PLUME runs. The black, blue and orange colors denote the runs for the $\mathrm{REF_{NO}}$, $\mathrm{REF_{WTG}}$ and $\mathrm{FAST_{WTG}}$ cases, respectively. Grey bands refer to the nighttime. The red dots in (a) indicate the inversion height in the climatology along the composite trajectory.

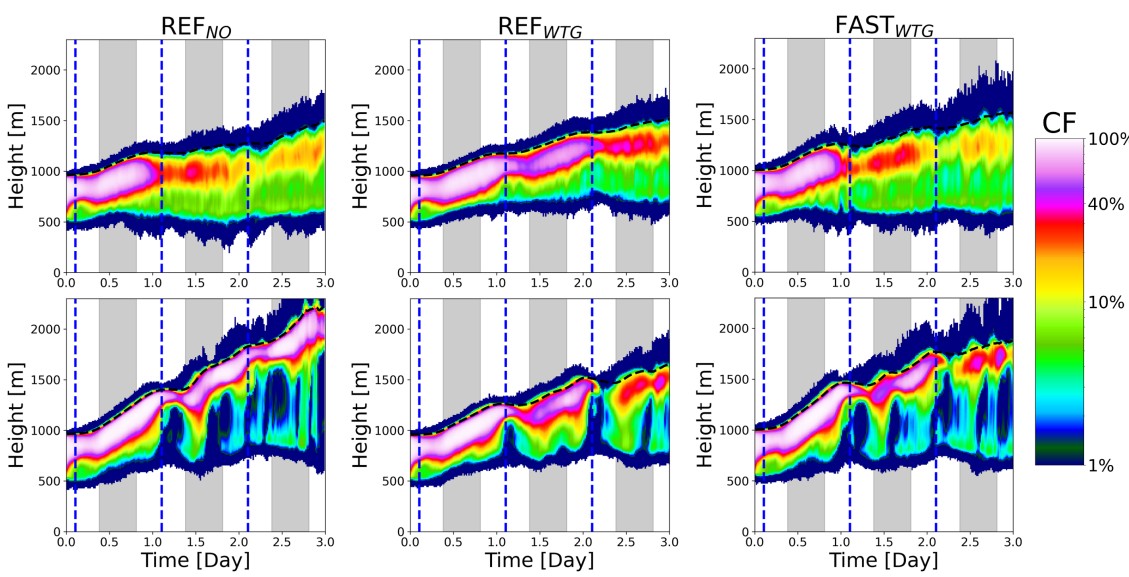

**Figure 3.** Time evolution of the vertical profiles in cloud fraction (CF). The left (a,d), middle (b,e) and right (c,f) columns are for the $REF_{NO}$, $REF_{WTG}$ and $FAST_{WTG}$ cases, respectively. The upper (a-c) and lower (d-f) rows represent the CTRL and PLUME runs, respectively. The black dashed line denotes the inversion height, and the blue dashed line represents the time of local noon.

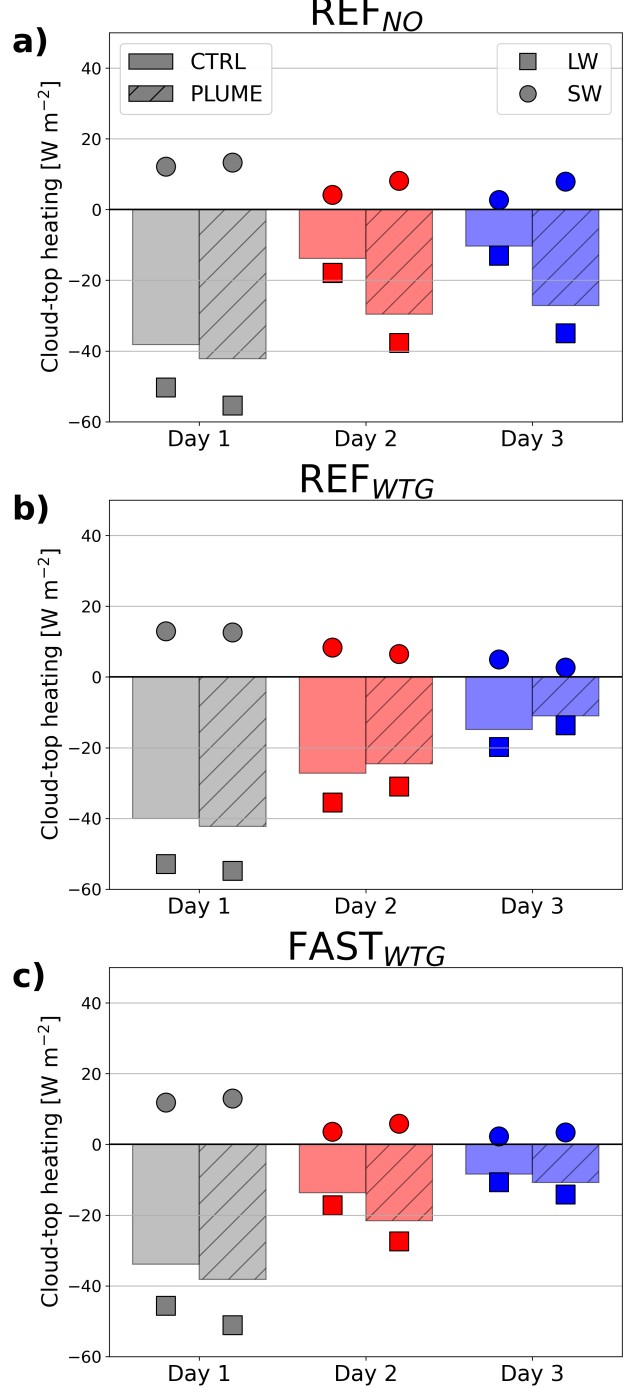

**Figure 4.** Cloud-top radiative heating rates in the CTRL and PLUME runs on Day 1, 2, and 3 for the (a) $REF_{NO}$, (b) $REF_{WTG}$, and (c) $FAST_{WTG}$ cases. Bars denote the net (LW+SW) cloud-top radiative heating rate. The smaller squares and circles indicate the average LW and SW radiative heating rates, respectively.

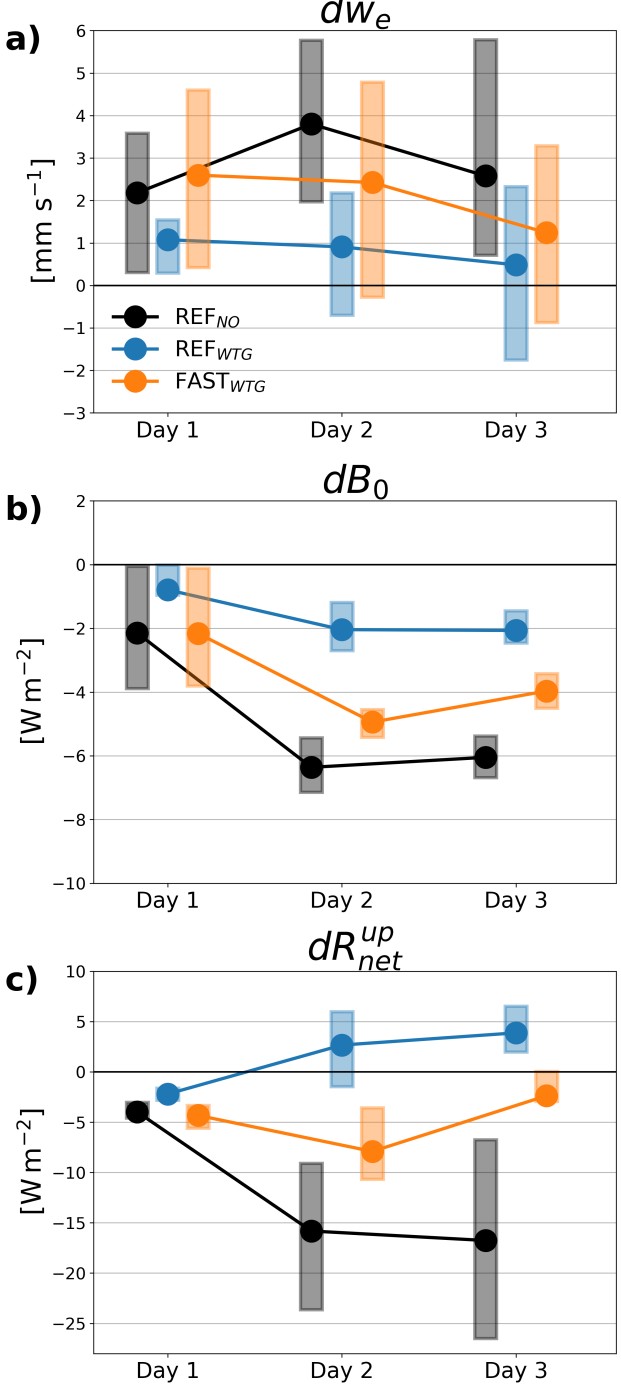

**Figure 5.** Changes in the (a) entrainment rate ($w_e$), (b) surface buoyancy flux ($B_0$) and (c) cloud radiative heating rate at the upper MBL ($dR_{net}^{up}$) caused by aerosol injections. Solid dots denote the diurnal means of the changes, while the shaded bars represent the interquartile range of the changes.

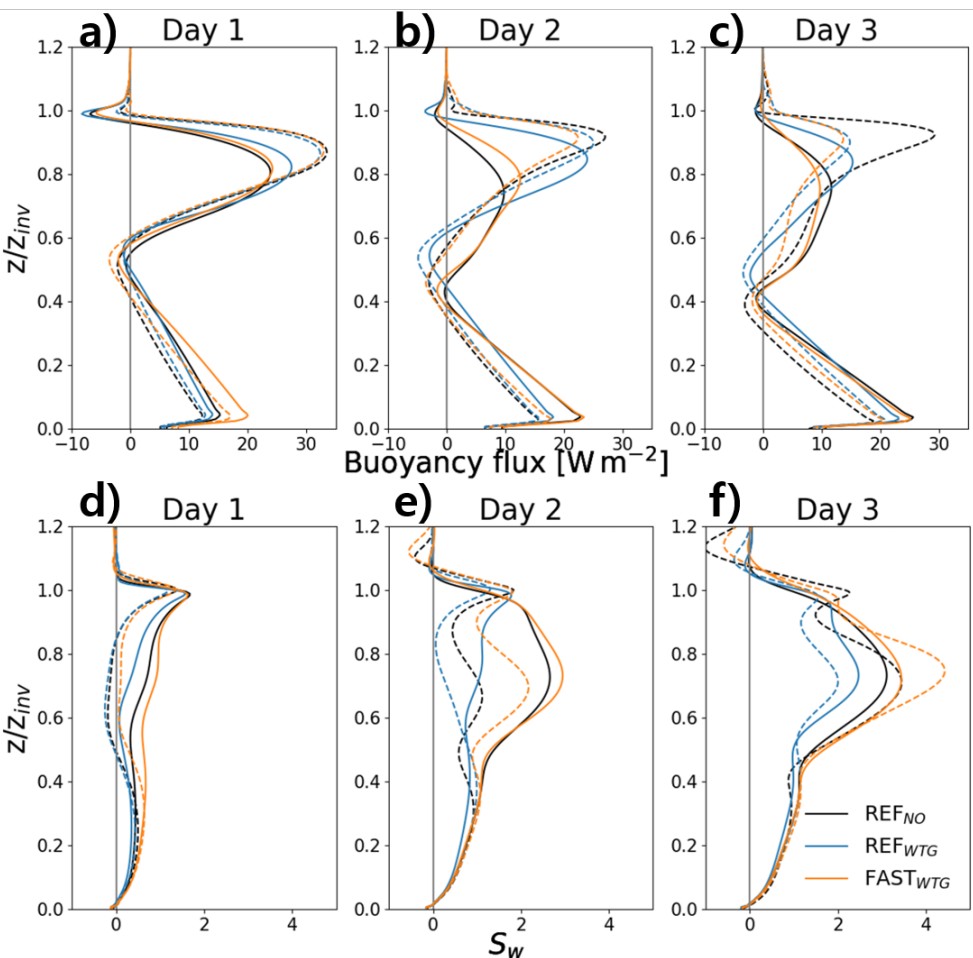

**Figure 6.** Diurnal-mean vertical profiles of (a-c) buoyancy flux and (d-f) skewness of the vertical wind speed on (a,d) Day 1, (b,e) Day 2 and (c,f) Day 3 cases. Black, red and blue lines denote the profiles of $REF_{NO}$, $REF_{WTG}$, and $FAST_{WTG}$ cases, respectively. Solid and dashed lines represent the profiles for the CTRL and PLUME runs, respectively.

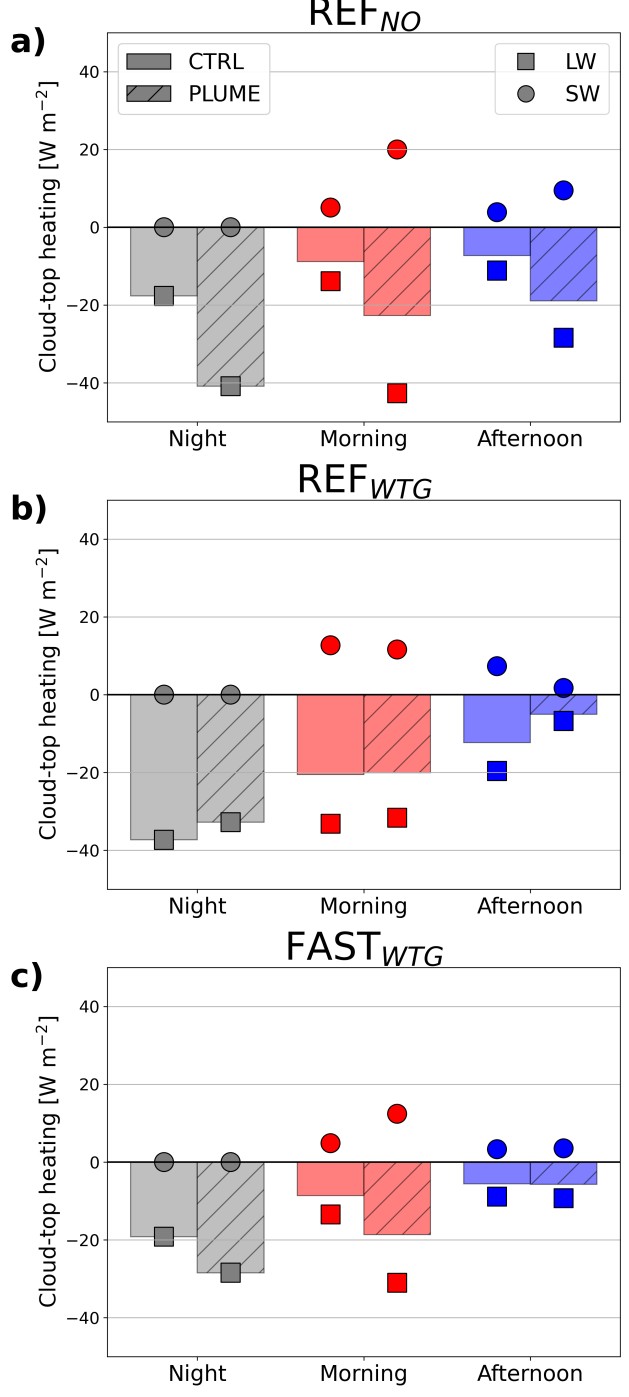

**Figure 7.** Same as Fig.4, but for different three time ranges (Night, Morning, and Afternoon), on Day 2.

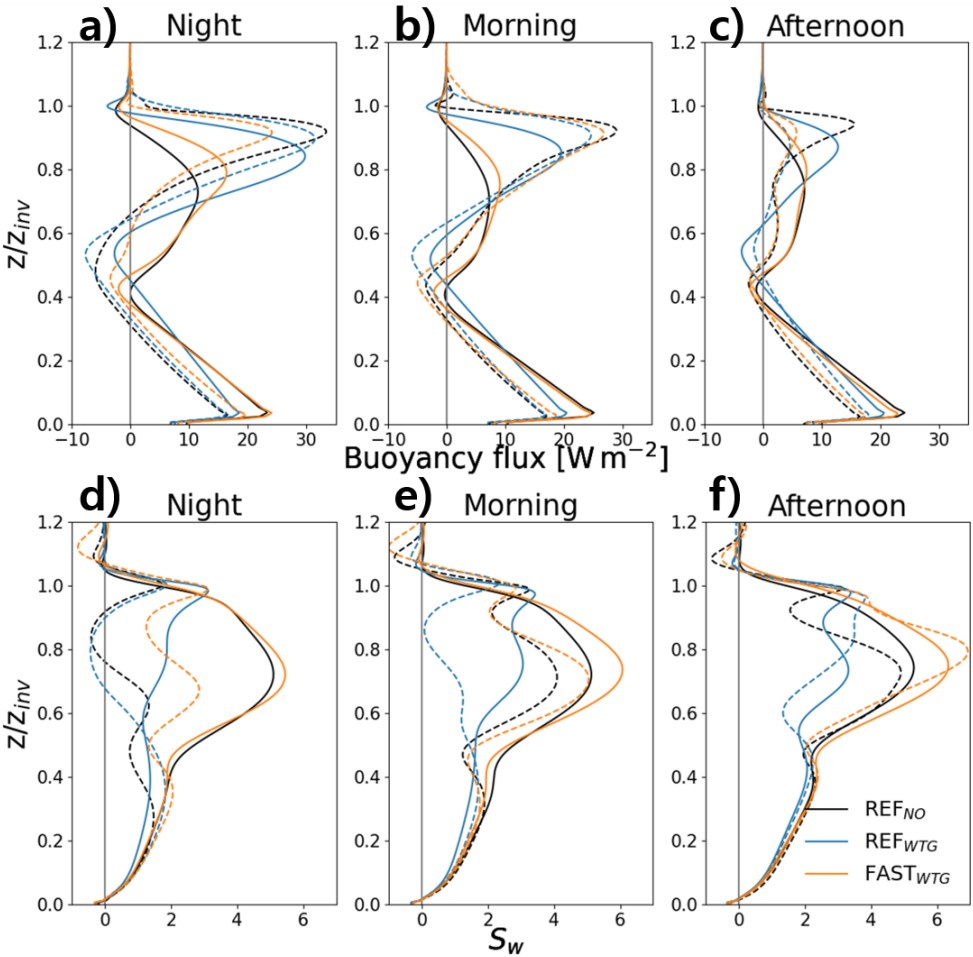

**Figure 8.** Same as Fig.6, but for different three time ranges (Night, Morning, and Afternoon), on Day 2.

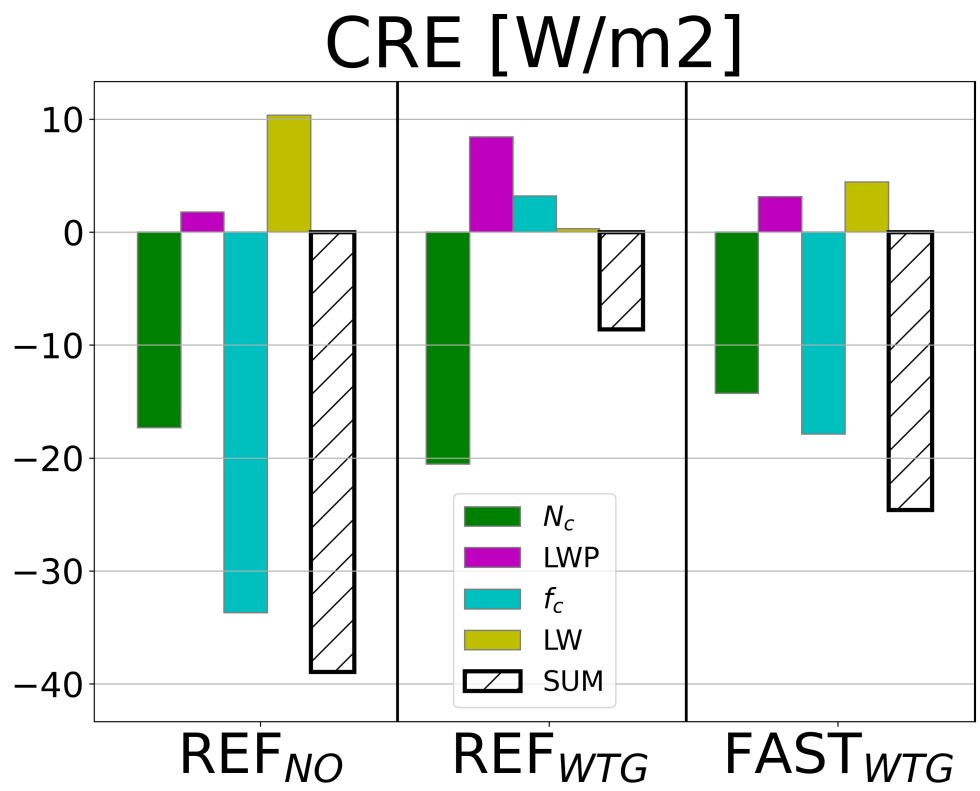

**Figure 9.** The difference in the decomposed 3-day cloud radiative effect between the PLUME and CTRL runs (dCRE) for each case, showing contributions from $\mathrm{dCRE}_{N_c}$ (green), $\mathrm{dCRE}_{\mathrm{LWP}}$ (magenta), and $\mathrm{dCRE}_{f_c}$ (blue) and $\mathrm{dCRE}_{\mathrm{LW}}$ (yellow). See Table 2 for the values on each day.

**Table 1.** Description of the cases analyzed in this study.

| Case Name | $REF_{NO}$ | $REF_{WTG}$ | $FAST_{WTG}$ | $REF_{weak}$ | $REF_{FT}$ | $REF_{MBL}$ |
|---|---|---|---|---|---|---|
| Case in Sandu and Stevens (2011) | REF | REF | FAST | REF | REF | REF |
| WTG | Off | On | On | On | On | On |
| Initial $N_a$ in MBL [cm$^{-3}$] | 33 | 33 | 33 | 33 | 33 | 300 |
| $N_a$ in FT [cm$^{-3}$] | 100 | 100 | 100 | 100 | 55 | 100 |
| Aerosol injection rate [$10^{16}$ particles s$^{-1}$] | 1.2 | 1.2 | 1.2 | 0.3 | 1.2 | 1.2 |

**Table 2.** Summary of diurnal-average and total changes in cloud radiative effect [W m$^{-2}$] for the PLUME runs relative to the CTRL runs.

| Case | | $dCRE_{N_c}$ | $dCRE_{LWP}$ | $dCRE_{f_c}$ | $dCRE_{LW}$ | $dCRE$ |
|---|---|---|---|---|---|---|
| $REF_{NO}$ | Day 1 | -19.8 | -1.0 | -10.8 | 2.8 | -28.7 |
| | Day 2 | -20.1 | 6.5 | -44.1 | 11.1 | -46.6 |
| | Day 3 | -12.1 | -0.2 | -46.2 | 17.1 | -41.4 |
| | Total | -17.3 | 1.8 | -33.7 | 10.3 | -38.9 |
| $REF_{WTG}$ | Day 1 | -17.5 | 2.4 | -2.6 | 1.2 | -16.4 |
| | Day 2 | -28.0 | 13.2 | 1.8 | 0.3 | -12.7 |
| | Day 3 | -16.1 | 9.7 | 10.4 | -0.6 | 3.4 |
| | Total | -20.5 | 8.4 | 3.2 | 0.3 | -8.5 |
| $FAST_{WTG}$ | Day 1 | -18.2 | 0.1 | -12.8 | 2.9 | -27.9 |
| | Day 2 | -15.6 | 6.8 | -30.3 | 6.4 | -32.6 |
| | Day 3 | -9.0 | 2.5 | -10.6 | 4.0 | -13.1 |
| | Total | -14.2 | 3.1 | -17.9 | 4.5 | -24.6 |

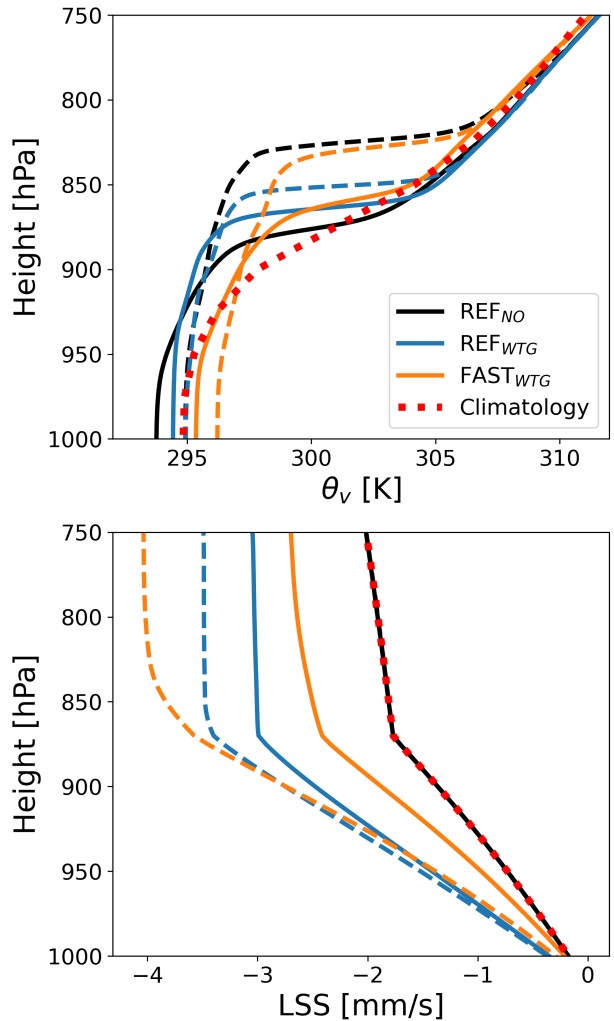

**Figure A1.** Vertical profiles of (a) virtual potential temperature $\theta_v$ and (b) large-sacle subsidence for the CTRL (solid) and PLUME (dashed) runs on Day 2 for the three cases. The red dotted line represent the ERA5 climatology at the location of climatological trajectory on Day 2 used for the background profile for the WTG.

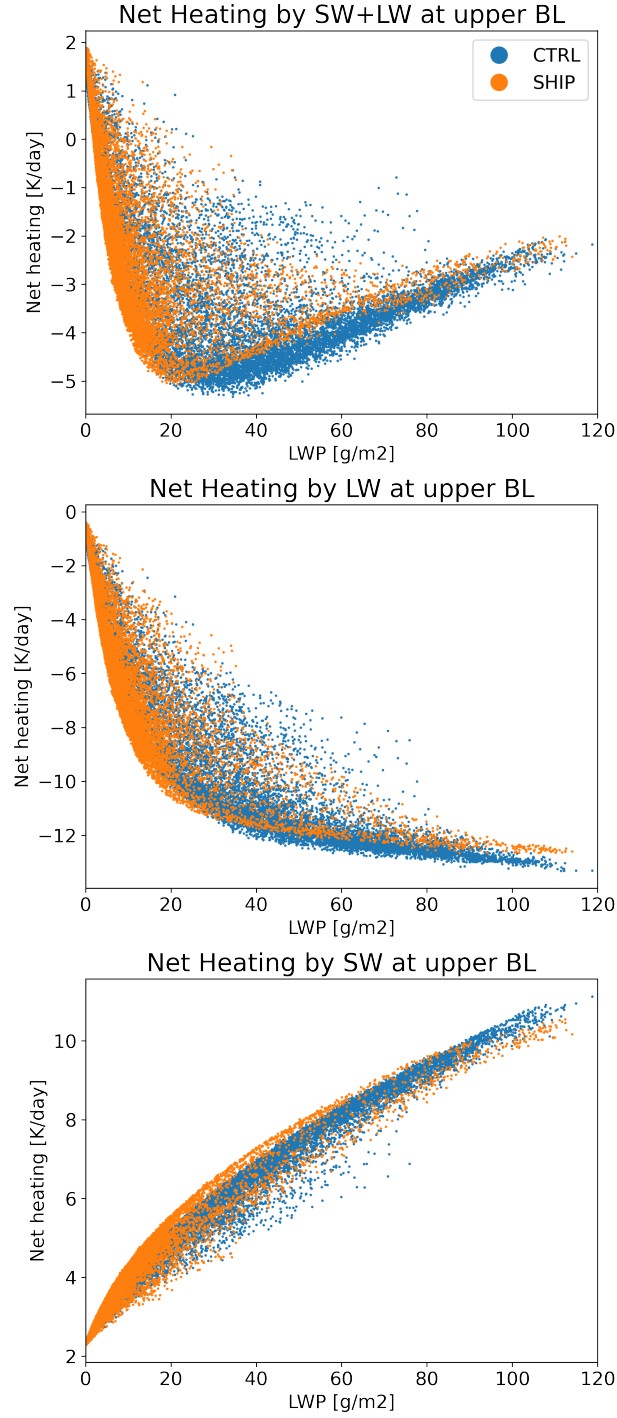

**Figure B1.** (a) SW and LW radiative heating rates by clouds in the upper part of BL (i.e., the mean at the levels in the upper half of the MBL) on Day 1.25 for the CTRL (blue) and PLUME (orange) runs. (b,c) same as (a), but for the LW and SW only, respectively.

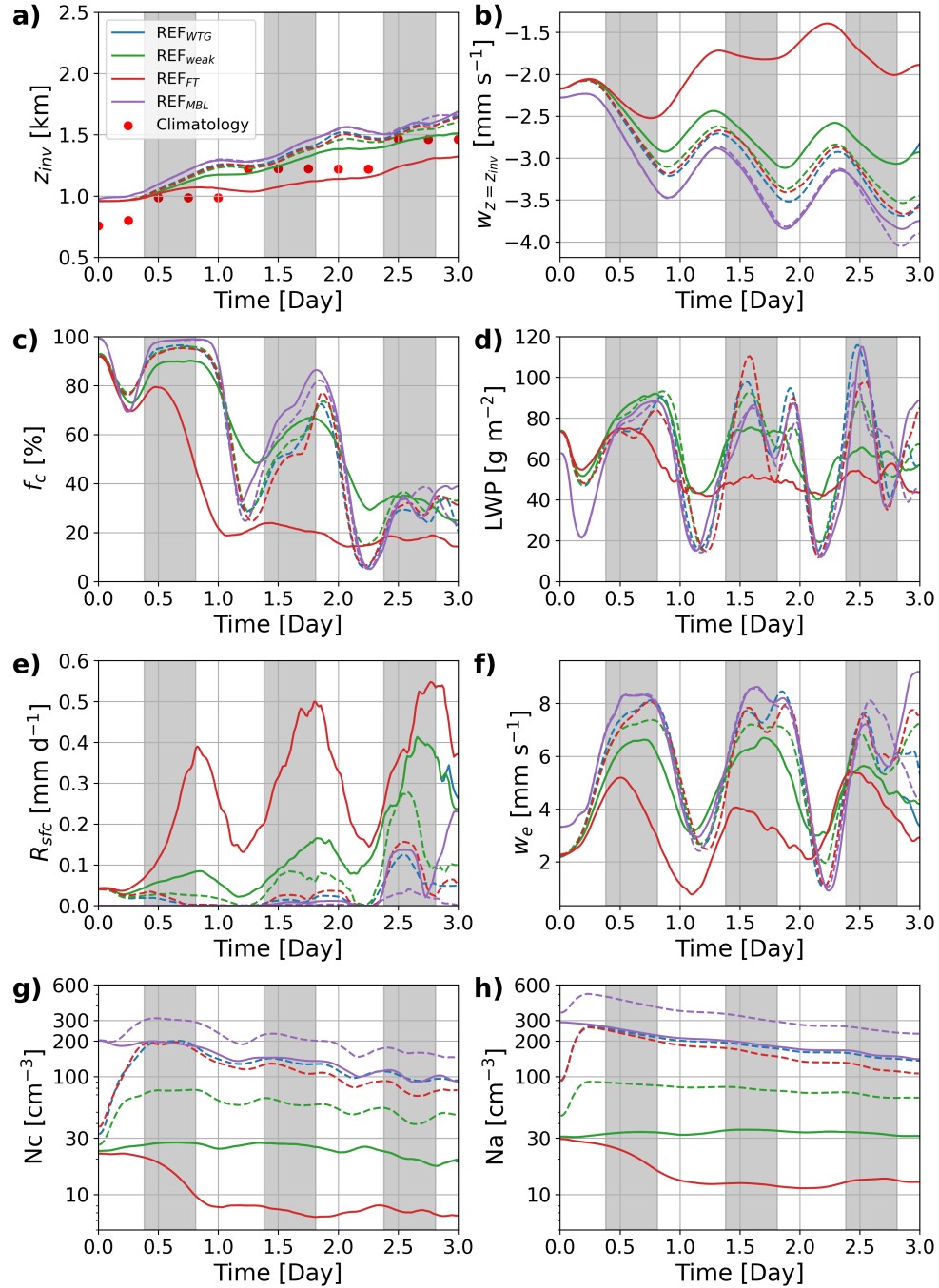

**Figure C1.** Same as Fig.1, but for the REF$_{\mathrm{WTG}}$ (blue), REF$_{\mathrm{weak}}$ (green), REF$_{\mathrm{FT}}$ (red) and REF$_{\mathrm{MBL}}$ (purple) cases.

**Table C1.** Same as Table 2, but for the $REF_{weak}$, $REF_{FT}$ and $REF_{MLB}$ cases

| Case | | $dCRE_{N_c}$ | $dCRE_{LWP}$ | $dCRE_{f_c}$ | $dCRE_{LW}$ | $dCRE$ |
|---|---|---|---|---|---|---|
| $REF_{weak}$ | Day 1 | -10.3 | 0.7 | -2.4 | 1.0 | -11.0 |
| | Day 2 | -13.9 | 10.7 | -0.9 | 0.2 | -3.9 |
| | Day 3 | -7.9 | 6.3 | 2.0 | 0.3 | 0.8 |
| | Total | -10.7 | 5.9 | -0.4 | 0.5 | -4.7 |
| $REF_{FT}$ | Day 1 | -16.1 | 1.3 | -20.0 | 3.9 | -30.8 |
| | Day 2 | -16.7 | 1.9 | -42.8 | 7.2 | -50.5 |
| | Day 3 | -11.8 | 0.8 | -8.8 | 4.1 | -15.6 |
| | Total | -14.9 | 1.3 | -23.9 | 5.1 | -32.3 |
| $REF_{MBL}$ | Day 1 | -3.7 | 0.2 | -0.1 | 0.0 | -4.0 |
| | Day 2 | -6.4 | 3.1 | 2.0 | -0.7 | -1.9 |
| | Day 3 | -4.0 | 5.2 | 2.8 | 0.1 | 4.0 |
| | Total | -4.7 | 2.7 | 1.6 | -0.2 | -0.6 |