# Peer review of "Impact on the stratocumulus-to-cumulus transition of the interaction of cloud microphysics and macrophysics with large-scale circulation"

_EGUsphere, 2024_

## Author Comment (AC1)

**Some general comments**

**The paper demonstrates that including the response of large-scale subsidence to changes in the mean vertical buoyancy profiles is crucial for assessing the strength of the cloud response to salt spraying in the stratocumulus to cumulus transition (SCT) regime. It may be helpful to give more attention to explaining the choices made in setting up the vertical profiles of aerosol concentrations (Na) according to Table 1, particularly since Na is relatively low in the boundary layer and higher in the free troposphere, which may be typical of the location studied in the paper. Additionally, as the study on the interaction between boundary layer turbulence and large-scale circulation is performed using a high-resolution large-eddy simulation model on a small domain, could the authors briefly discuss the pros and cons of this approach compared to using a global model, such as the one applied by Wan et al. (Nature, 2024)?**

Typical values of accumulation mode aerosol number concentration over the NE Pacific have been reported in previous papers (e.g., Mohrmann et al.,2018). We have added some sentences in the second paragraph in the Section 2.3 including reference papers, explaining the background for the choice of values:
*"Na in the MBL and FT are 33 and 100 cm−3, respectively, for all the three cases. We use an Na in theMBL (i.e., 33 cm−3) that is lower than its climatological mean value (greater than ~ 100 cm−3) in order to produce precipitation significant enough to be influenced by aerosol injections, while the chosen FT Na is kept at its climatological mean value (e.g., Mohrmann et al., 2018)."*

The reason for not using a global-scale model (as in Wan et al., Nature, 2024) is given on lines 97-100, i.e.: "Larger-scale and possible global simulations might be required to understand such coupling, though such models struggle to resolve the small-scale processes driving cloud adjustments to aerosol perturbations." To be even clearer, we have edited this to read:
"Larger-scale (e.g. regional to global-scale) simulations will explicitly simulate these large-scale circulation responses to aerosol perturbations and cloud brightening. However, these larger-scale models can not resolve the small-scale processes driving cloud adjustments to perturbations, and as such will not capture the *coupling* between cloud responses and the large-circulation change, and the resulting feedbacks to cloud brightening."

The limits of the use of the WTG in LES are noted in the Discussion Section (from Line 441 to 454). Global climate models, with a longer time and larger domain enough to allow the dynamic adjustment in large-scale circulation, can represent the changes in thermodynamic status in the free troposphere, which can in-turn influence aerosol-cloud interactions. We illustrate the importance of using a global climate model to represent the large-scale dynamic adjustment in the second paragraph from the end of the Introduction Section.

**(Minor) remarks**

**line 117: Include a reference to the Morrison microphysics scheme?**

The citation for the Morrison microphysics scheme is added to Line 119.

**line 135: The vertical profiles of the buoyancy do generally depend on time and space, and according to the Lagrangian setup of the simulations, the position of the LES domain changes with time. However, it is not immediately clear whether the 'diagnosed buoyancy profiles' are taken constant during the simulation, or that they depend on the position (and local time) of the LES domain? The results suggest the latter, in particular the caption of Fig. A1 states that "The red line represent(s) the ERA5 climatology at the location of climatological trajectory on Day 2" (note that Fig A1b does not show a red line). All the relevant details on the WTG application are present, but I find them a bit scattered around in the manuscript. As a side note, the main text mentions "domain-mean anomalies of virtual temperature and diabatic heating with respect to diagnosed buoyancy profiles", while Appendix A uses the virtual potential temperature. Perhaps include their definitions?**

The 'diagnosed buoyancy profile' in Line136 is derived from the climatological mean profile at grid points along the trajectory (as given by Table C1 in Sandu et al. 2010) in ERA5. Thus, the buoyancy profiles change depending on the position (and local time), not in the LES domain but in ERA5 mean profile at the locations of the trajectory given by Table C1 in Sandu et al. (2010). Since the large-scale subsidence from climatology (red) is exactly overlapped with the REF_NO (because there is no adjustment in LSS), the red line in Fig.A1b does not seem to be drawn, but does actually exist. I have made the redlines in Fig.A1 thicker, so that they look apparent.

For better clarity in the main text, the sentence on line 135 has been edited to read:

"The basic principle of the WTG approximation is that domain-mean anomalies of virtual temperature in the simulated column are calculated relative to location- and time-dependent climatological buoyancy profiles, and these are used as the primary drivers of the perturbation in the column mean vertical motion."

**Line 247, the buoyancy flux used is different from its definition $\langle w'b' \rangle = g/\langle w'\theta_v' \rangle$ (with $\langle \rangle$ indicating a slab mean value). The statement that "changes in SHF play a leading role in changes in the surface buoyancy flux in these simulations" seems a bit too simplistic. The SHF and LHF values in the Sandu cases are on the order of 10 and 100-200 W/m2, respectively (see Sandu and Stevens 2011). Because the LHF can be an order of magnitude larger than the SHF, the term 0.07 x LHF can become as large as the SHF.**

I agree with the fact that the typical order of magnitude in LHF is much greater than that in LHF. However, what is relevant here is not the values themselves, but their changes due to the aerosol perturbation. Because the order of magnitude of the changes of SHF and LHF are similar, the contribution of the *change* in SHF to the surface buoyancy flux is much greater than for that of the LHF. To make this clear, we have added "Since the order of magnitude of the changes in SHF and LHF from the aerosol injections are similar and … (etc) " to  the sentence in Line 281 to make it clearer.

**2.3 Data. N_a is not defined (first occurrence in the text, line 156). Could the initial values for N_a in the MBL and FT presented in Table 1 be motivated? In particular the lower values of N_a in the MBL compared to the ones in the free troposphere?**

Thank you for pointing this out. The N_a definition is given when first mentioned. We have added sentences in the Section 2.3 to state the motivation of initial values for Na in the MBL and FT.

**Line 157, Spin up procedure. An 18 hour spin-up period is applied. Can some more details be given about how this is done? For example, are the large-scale conditions set constant or not during spin up, is nudging applied?**

We have added a sentence to give more detail:
*"To allow the MBL and clouds to sufficiently evolve, the runs are spun up for 18 hours nudged to initial profiles with a timescale of 10 minutes. A long spin-up time is chosen to allow the mesoscale organization to fully develop, since it is important for determining cloud adjustments (e.g., through precipitation). Throughout the simulation, temperature and specific humidity 500 m above inversion are nudged with a timescale of one hour to the climatological mean profiles along the trajectory. After the spin-up, ~"*

**Line 158, can the choice of the aerosol injection rate be motivated? Does it involve evaporative cooling of the sprayed water?**

In the previous papers, the injection rates typically ranges from order of 10^15 to 10^17 (See Table 2 in Wood 2021). Also, the two representative MCB scenarios given in Wood 2021 also has injection rates of 6x10^16 and 6x10^15. Thus, we think that our injection rate (1.2x10^16) is likely reasonable. The evaporative cooling of the sprayed water is not considered, but recent outdoor studies have suggested that this may be less of an issue than has been previously hypothesized (Hernandez-Jaramillo et al., 2024).

**Line 163, rotation of the domain. Due to surface friction and the resulting momentum fluxes the wind in the boundary layer will turn with respect to the geostrophic wind direction. However, it is stated that (with a rotated domain) 'the x component of background wind velocity is approximately zero', which suggests that the ageostrophic component of the wind vector is close to zero. Please clarify.**

The rotation of the domain makes x-component velocity of geostrophic wind approximately zero. However, x-component velocity of ageostrophic component should not be zero. Also, as you mentioned, the wind turns with height before reaching a roughly steady direction higher in the MBL due to surface friction.

To clarify, we have edited the sentence as:
*"The domain is rotated to align the geostrophic background wind with the y-axis (i.e., wind in the lower MBL turns with height and reaches a steady direction to nearly y-axis in the upper MBL) to minimize the advection of plume in the x direction (Figure~\ref{fig01})."*

**Line 177, Figs 2 and 3, f_c is defined as a cloud fraction but also as a cloud cover. The cloud fraction is often defined as the ratio of cloudy area to the total area on a horizontal plane as is applied in Fig. 3, but Fig. 2 shows the cloud cover (ratio of cloudy air columns to all vertical columns).**

Thank you for pointing them out. We have changed "cloud fraction" to "cloud cover", except the cases relevant to Fig.3.

**Line 188, 'intensified subsidence .. delivers more CCN into the MBL' . Aren't CCN entrained into the MBL, subsidence is just pushing down the boundary layer?**

Thanks for catching this. We have edited the sentence to:

"*when intensified subsidence is accompanied by enhanced entrainment, more CCN are incorporated into the MBL.*"

**Line 279, 'turbulence dissipation by the decreased surface buoyancy flux'. The sign of the buoyancy flux in the subcloud layer matters in this respect, at heights where it is negative it will tend to diminish turbulent kinetic energy.**

We agree that the absolute value of the buoyancy flux is important, but this text is focusing on changes rather than absolute values. We have edited lines 277-281 in an effort to improve their clarity: "Among the three cases, the decrease in $B_0$ is greatest for REF_NO. However, the weaker driving of turbulence by the smaller surface buoyancy flux ($dB_0 \sim -6$ Wm-2 on day 3) is more than offset by stronger radiative cooling in the upper part of the MBL ($dR^{up}\_net \sim -15$ Wm−2 on Day 3). As a result, the increased turbulence in the MBL is intense enough to sustain the stratocumulus layer."

**Line 286, Can you explain what is meant with 'the aerosol-cloud interaction is not yet saturated'?**

"Saturated" here indicates that the impact of aerosol injections on the cloud reaches the adjustment equilibrium state.

We have rephrased this to read: "On Day 1, $dw_e$ increases with time as the plume track spreads, and as the cloud adjustments to the aerosol injection have not yet reached equilibrium.", and added some reference papers dealing with the time scales (Schubert et al.,1979, Wood 2007 and Glassmeier et al., 2021).

**Appendix A. The WTG application is strongly based on ERA5 fields. Given the typical biases in weather and climate models of various quantities in the SCT regime, it is maybe worthwhile to briefly discuss the accuracy of the vertical profiles of the thermodynamic variables in the SCT regime in ERA5?**

To address the mentioned issue, I have added some sentences to the last paragraph of the Discussion Section:
*Additionally, the WTG approach in this study relies on thermodynamic profiles from ERA5 reanalysis, which, though useful, may introduce biases typical of weather and climate models in the SCT regime. ERA5 vertical profiles of temperature, moisture, and other variables may not fully capture the subtle thermodynamic gradients and*

*interactions characteristic of SCT regions, potentially affecting the representation of cloud formation and dissipation processes.*

**Fig. 2: refer to 'large-scale' vertical velocity in Fig. 2b. Explain the meaning of the grey bands?**

I appreciate your pointing it out. I added a sentence to the caption in Fig.2: "Grey bands refer to the nighttime"

**Typos**

**line 105: "Arkawa"**

I have edited it to "Arakawa".

**RC#2 Comments:**

1. Lines 70-75: Does Dagan 2022 quantify the time scale for this response of the large-scale circulation?

The paper does not explicitly quantify a timescale for the response of the large-scale circulation. It primarily describes the dynamics using a weak temperature gradient (WTG) approximation and focuses on changes in large-scale vertical velocities, temperature, and moisture advections in response to aerosol perturbations. The timescale of changes is more implicitly addressed through the behavior of the large-scale forcing (LSF) variables, which stabilize after the first two days of simulation.

2. Line 114: \approx might be better than \sim. Please check.

I have edited \sim to \approx

3. The description of WTG is very poor. Not enough details are provided except for a reference to Blossey et al 2009. Considering the novelty of the topic it would be good to have a nice description of WTG (with equations) and the values for the parameters used in this study. The focus should be on providing enough information so that someone who does not know about WTG knows enough to follow the paper. Currently, WTG seems like a black box switch that was turned on and that's it.

What equation was solved? My guess is some kind of 2D wave wave equation. What is the physical meaning of the terms in the equation? Over what timescales do they operate?

The authors should use sec 2.1 as reference. SAM is a well documented model, yet the authors have provided enough details about the model in Sec 2.1. Please provide more details about WTG.

Many more details about WTG have been added, including the wave equation resolved in the scheme and the parameters we used. In addition, we note that the appendix of Blossey et al. (2009) provides a self-contained description of the WTG method for interested readers.

4. Line 145: please state the months

We have added [June-August (JJA)] when JJA is first mentioned, and (JJA) to the line.

5. Line 151: What about the subsidence corrections given in Bretherton Blossey 2014? This was used in Yamaguchi et al 2017 and Prabhakaran et al 2024. Any comments?

We use the corrected subsidence given in Bretherton and Blossey (2014). I appreciate your correction. We have changed the first sentence in Section 2.3 to mention the use of corrected subsidence and Sandu and Stevens (2011) to Bretherton and Blossey (2014) in Line 151.

6. How reliable are the buoyancy profiles in ERA5 near the inversion? It is not clear to me why in the CTRL case WTG is required. See the correction to subsidence provided in Bretherton Blossey 2014. Wouldn't this address the concerns associated with buoyancy anomalies? Additionally, the buoyancy anomalies could also be an artifact of the surface flux parameterization and not related to subsidence. Any comments about this?

Temperature and moisture profiles (and thus, buoyancy profiles) in ERA5 are generally considered reliable, though the coarse vertical resolution near the inversion layer can introduce limitations. The use of WTG in the CTRL runs is motivated by the fact that the Sandu and Stevens (2011) case study was not designed to be consistent with climatology along the trajectory.  The trajectories used to develop the case study were selected "based on the trajectories that are the most likely to experience such a transition."  (Quote from Sandu et al., 2010, https://doi.org/10.5194/acp-10-2377-2010, sec. 2.3 "Conditional sampling").  As a result, even though buoyancy anomalies are minimal in the CTRL runs, the slight adjustments in subsidence made by WTG can influence CCN incorporation into the boundary layer, which in turn significantly impacts cloud properties. Figure 2 illustrates that cloud properties (e.g., precipitation rate, inversion height, cloud fraction, LWP) differ noticeably between the REF_NO and REF_WTG CTRL runs. Surface fluxes may contribute to buoyancy anomalies secondarily, as variations in surface precipitation due to WTG can alter surface fluxes. I have added additional clarifying sentences on this point as follows:

*Although subsidence correction by Bretherton and Blossey (2014) does reduce buoyancy anomalies, small anomalies can still persist. Consequently, the implementation of WTG induces a minor change in subsidence within the simulation, leading to variations in cloud and MBL properties. This rationale underlies the necessity of running simulations without aerosol injection for both REF_NO and REF_WTG cases to establish a clear baseline for comparing background and perturbed conditions.*

7. How does the subsidence change with time? Wouldn't the dilution of aerosol due to lateral spreading weaken the changes in subsidence? And how do you justify the uniform changes in subsidence across the domain? On Day 1, the aerosol is still spreading laterally.

It might be useful for us to respond to these individual questions out of order.

First, "And how do you justify the uniform changes in subsidence across the domain?"

Local differences in entrainment between the plume and its environment can be represented by compensating circulations between the boundary layer and lower free troposphere.  This process is illustrated, for example, in figure 18 of Bretherton et al (2010, https://doi.org/10.3894/JAMES.2010.2.14), where "POC" would correspond to the environment in our simulations and "Overcast" would correspond to the plume.  Within

the simulation domain, these resolved circulations compensate for the differing entrainment rates in these regions and maintain an approximately $z_{inv}$ (inversion height) across the domain. However, the domain-mean inversion height may drift from the climatological value over time and create an inconsistency between the model's buoyancy profile and the climatological one. The WTG method will act to limit this inconsistency in the domain-mean buoyancy profile, with increases in subsidence that are roughly proportional to the mismatch between the model's $z_{inv}$ and that implied by the climatological profile.

Second, "How does the subsidence change with time?"

The WTG-induced subsidence responds to the buoyancy anomaly between the model's domain mean profile and that of the climatology. Both of these profiles are evolving over time along the transition, so it is expected that the WTG-induced subsidence will as well. As noted above larger mismatches in inversion height between the model and climatology will lead to stronger subsidence changes.

The subsidence evolution at inversion height with time is shown in Figure 2b, so the changes in subsidence in these time series reflect both the changes in subsidence with height as the inversion rises and the changes due to the WTG-induced subsidence. The diurnal variations in subsidence in the WTG simulations reflect, in part, the way the WTG-induced subsidence is responding to diurnal variations of cloud-top entrainment.

Last, "Wouldn't the dilution of aerosol due to lateral spreading weaken the changes in subsidence?"

Lateral spreading of the plume results in a change from a narrower-but-more-polluted to a wider-but-less-polluted plume without a change in the total number of injected aerosols. As the spreading plume still includes enough aerosols to impact the surface precipitation (which falls to near zero in both WTG runs with aerosol injection; see Figure 2e), we could expect the spreading plume to induce a broader area of non-precipitating and more strongly entraining cloud, since the initially narrow plume had much more aerosol than required for precipitation suppression. As a result, one might expect the entrainment rate to rise as the plume spreads (mostly over the first day as seen in figure 1), and this is consistent with the growing entrainment rates of the REF_WTG and FAST_WTG PLUME runs during the first day.

8. Lines 247-248: LHF>>SHF, so changes in LHF is also important. How do you explain this?

This comment duplicates the third minor comment from RC#1, so please refer to my response. I have edited the lines to make my argument clearer.

9. Lines 397: Please include more references, not just Chun et al 2023.

We have added Albrecht 1989 for precipitation suppression, and Chen et al 2024 for surface flux impacts.

10. Can you comment on the limitations of WTG? Dagan et al 2022 raised critical points regarding the results in ABott & Cronin. A brief discussion about this would improve the paper further.

We have edited the fourth paragraph in the Discussion Section.

**References**

C. Hernandez-Jaramillo, D., Medcraft, C., Campos Braga, R., Butcherine, P., Doss, A., Kelaher, B., Rosenfeld, D., & P. Harrison, D. (2024). New airborne research facility observes sensitivity of cumulus cloud microphysical properties to aerosol regime over the great barrier reef. Environmental Science: Atmospheres, 4(8), 861–871. https://doi.org/10.1039/D4EA00009A

Mohrmann, J., Wood, R., McGibbon, J., Eastman, R., & Luke, E. (2018). Drivers of seasonal variability in marine boundary layer aerosol number concentration investigated using a steady state approach. *Journal of Geophysical Research: Atmospheres*, *123*(2), 1097-1112.

Wan, J.S., Chen, C.C.J., Tilmes, S., Luongo, M.T., Richter, J.H. and Ricke, K., 2024. Diminished efficacy of regional marine cloud brightening in a warmer world. *Nature Climate Change*, pp.1-7.

Wood, R. (2007). Cancellation of Aerosol Indirect Effects in Marine Stratocumulus through Cloud Thinning. Journal of the Atmospheric Sciences, 64(7), 2657–2669. https://doi.org/10.1175/JAS3942.1